# MAPK-dependent hormonal signaling plasticity contributes to overcoming *Bacillus thuringiensis* toxin action in an insect host

Zhaojiang Guo [1,4✉], Shi Kang[1,4], Dan Sun[1,4], Lijun Gong[1], Junlei Zhou[1], Jianying Qin[1], Le Guo[1], Liuhong Zhu[1], Yang Bai[1], Fan Ye[1], Qingjun Wu[1], Shaoli Wang[1], Neil Crickmore [2], Xuguo Zhou [3] & Youjun Zhang [1✉]

The arms race between entomopathogenic bacteria and their insect hosts is an excellent model for decoding the intricate coevolutionary processes of host-pathogen interaction. Here, we demonstrate that the MAPK signaling pathway is a general switch to *trans*-regulate differential expression of aminopeptidase N and other midgut genes in an insect host, diamondback moth (*Plutella xylostella*), thereby countering the virulence effect of *Bacillus thuringiensis* (Bt) toxins. Moreover, the MAPK cascade is activated and fine-tuned by the crosstalk between two major insect hormones, 20-hydroxyecdysone (20E) and juvenile hormone (JH) to elicit an important physiological response (i.e. Bt resistance) without incurring the significant fitness costs often associated with pathogen resistance. Hormones are well known to orchestrate physiological trade-offs in a wide variety of organisms, and our work decodes a hitherto undescribed function of these classic hormones and suggests that hormonal signaling plasticity is a general cross-kingdom strategy to fend off pathogens.

[1] Department of Plant Protection, Institute of Vegetables and Flowers, Chinese Academy of Agricultural Sciences, Beijing 100081, China. [2] School of Life Sciences, University of Sussex, Brighton BN1 9QE, UK. [3] Department of Entomology, University of Kentucky, Lexington, KY 40546-0091, USA. [4] These authors contributed equally: Zhaojiang Guo, Shi Kang, Dan Sun. ✉email: guozhaojiang@caas.cn; zhangyoujun@caas.cn

nsects are the most speciose group of animals, with approximately 5.5 million species estimated to exist on earth[1]. Many insects are agricultural pests regularly causing 10–20% crop losses and with on-going global warming, they post a real threat to world food security[2]. The gram-positive bacterium *Bacillus thuringiensis* (Bt) can produce protein toxins to kill different insects with high host specificity and environmental safety[3], which makes it the most successful biopesticide for the last few decades[4]. Transgenic crops expressing Bt toxins (Bt crops) have become the cornerstone of bioinspired pest control technology, with >100 million hectares planted globally in 2018[5]. Although Bt products have provided unprecedented economic, environmental, and social benefits[3,6–10], the rapid evolution of Bt resistance in at least nine insect species in the field has seriously eroded their potential[4,11–14]. Unraveling the molecular mechanisms of Bt resistance has important implications for the sustainable utilization of Bt-based technology[15–17].

Bt Cry toxins exert toxicity in insect larval midguts via a multi-step process requiring protoxin activation, toxin–receptor interaction, toxin oligomerization, membrane insertion, and pore formation[18,19]. Alterations of midgut receptors such as cadherin (CAD), aminopeptidase N (APN), alkaline phosphatase (ALP), and ABC transporters (e.g., ABCC2) disrupt toxin binding and are generally associated with high-level resistance to Bt Cry toxins in insects[20,21]. The diamondback moth, *Plutella xylostella* (L.), is one of the most devastating and cosmopolitan agricultural pests[22]. It was the first insect to develop field-evolved resistance to Bt biopesticides[23], and the availability of complete whole genome information[24] renders it an excellent model to probe how insect hosts withstand Bt infection during host–pathogen interaction. Previously, field-evolved resistance to Bt Cry1Ac toxin in *P. xylostella* has been linked to both a *cis*-mutation in the *PxABCC2* gene[25] and MAPK-mediated differential expression of *PxmALP*, *PxABCB1*, *PxABCC1-3*, and *PxABCG1* genes[26–28]. Although we found that the MAPK signaling pathway can alter the expression of multiple midgut genes related to Cry1Ac resistance in *P. xylostella*, its downstream response gene repertoires and upstream activation signals remained mysterious[29].

APN proteins are a class of endoproteases catalyzing the cleavage of neutral amino acids from the N-terminus of protein or polypeptide substrates[30]. They belong to the M1 family (metallo-type) of zinc-dependent aminopeptidases that are implicated in many physiological processes of diverse organisms[31]. In the insect midgut, besides their important role in food digestion, APNs were the first identified functional receptors of Bt Cry toxins[20,30]. Moreover, we had previously detected several differentially expressed midgut PxAPN genes in Bt Cry1Ac-resistant *P. xylostella*[32,33]. However, whether the differential expression of these PxAPN genes associated with Cry1Ac resistance was also *trans*-regulated by the MAPK cascade in *P. xylostella* was unclear.

Insect endocrinologists have studied insect hormones for more than a century, and they have discovered that two major insect hormones, juvenile hormone (JH) and 20-hydroxyecdysone (20E), act antagonistically with each other to coordinately orchestrate insect life-history traits including growth, development, and reproduction[34–37]. Moreover, JH and 20E are multifunctional players that can also participate in insect immune defense to pathogenic infection[38,39], and the MAPK signaling pathway is involved in this pleiotropic hormone signaling network[35,40]. Since exogenous hormone treatments can alter APN gene expression in insects[41], we also wanted to test whether altered levels of insect hormones can activate the MAPK cascades thereby *trans*-regulating the differential expression of PxAPN and potentially other midgut genes to confer Cry1Ac resistance in *P. xylostella*.

In this study, we further probe the mechanism of Bt Cry toxin resistance in *P. xylostella*, starting by looking at the potential role of APN genes, and we confirm that MAPK-mediated differential expression of APN and other midgut genes does lead to Cry1Ac resistance in *P. xylostella*. More importantly, we uncover that the MAPK cascade is activated and modulated by an enhanced pleiotropic hormone signaling pathway. This study provides a model for understanding how intracellular signaling networks shape the expression landscape of midgut genes causing Bt resistance/tolerance in insects within the context of balancing growth-defense tradeoffs.

## Results

**Genome-wide analysis of APN gene family in *P. xylostella*.** Based on currently available transcriptome and genome databases of *P. xylostella*, 18 M1 aminopeptidase genes including 15 APN genes were identified in silico, and their full-length cDNA sequences were successfully cloned except for *PxAPN3b* (Supplementary Table 1 and Supplementary Fig. 1a). A representative lepidopteran APN protein contains six common features (Supplementary Fig. 1b), including the characteristic gluzincin aminopeptidase motif GAMEN and the zinc-binding/gluzincin motif HEX2HX18E located in the peptidase_M1 domain which are conserved in nearly all of these M1 aminopeptidases (Supplementary Table 1 and Supplementary Fig. 1c). We found that the APN1-12 gene cluster possesses highly conserved synteny in both gene order and orientation in different lepidopteran insects, indicating that it has undergone tandem gene duplication during insect genome evolution (Fig. 1a). Although the paralogous PxAPN1-12 genes show similar features including exon number, size, and intron phase (Supplementary Fig. 1d), they share relatively low protein sequence similarity (Supplementary Fig. 1e), implying their evolutionary and functional diversity. A model-based phylogenetic analysis demonstrates that lepidopteran APN proteins cluster into 13 classes and are evolutionarily conserved in each class. Sister phylogenetic relationships were also observed between APN1 and APN3 and between APN5 and APN6, suggesting close protein structure and functional similarities within these pairs (Supplementary Table 2 and Fig. 1b).

**Altered expression of four PxAPN genes in resistant strains.** A previous study showed that *cis*-mutations in PxAPN genes are not linked to Cry1Ac resistance in *P. xylostella*[42]. Likewise, we did not detect mutations in PxAPN genes in any of our resistant *P. xylostella* strains. Nevertheless, considering that our previous *P. xylostella* midgut transcriptome and RNA-Seq studies had identified differentially expressed PxAPN genes in the resistant strains[32,33], the transcription profiles of these genes were further investigated. Unsurprisingly, given their function as digestive enzymes, the APN genes were found to be most highly expressed in the midgut and during the larval stages. Some (PxAPN1-9) were expressed to a greater extent than the others (Supplementary Fig. 2a). These data were compared with our previous midgut transcriptome[32] and RNA-Seq[33] data (Supplementary Fig. 2b), which showed that the PxAPN1-9 genes (except for *PxAPN7*) were expressed in midgut tissues of third-instar Bt susceptible larvae. When the constitutive midgut transcription profiles of these genes in fourth-instar larvae from both Bt-susceptible and -resistant strains were compared, it was revealed that both *PxAPN1* and *PxAPN3a* genes were substantially decreased in all of the Bt-resistant strains, whereas *PxAPN5* and *PxAPN6* genes were increased (Fig. 1c). Assessment of protein levels confirmed that PxAPN1 and PxAPN3a were both reduced in midgut samples from all the Bt-resistant fourth-instar larvae (Supplementary Table 3 and Fig. 1d).

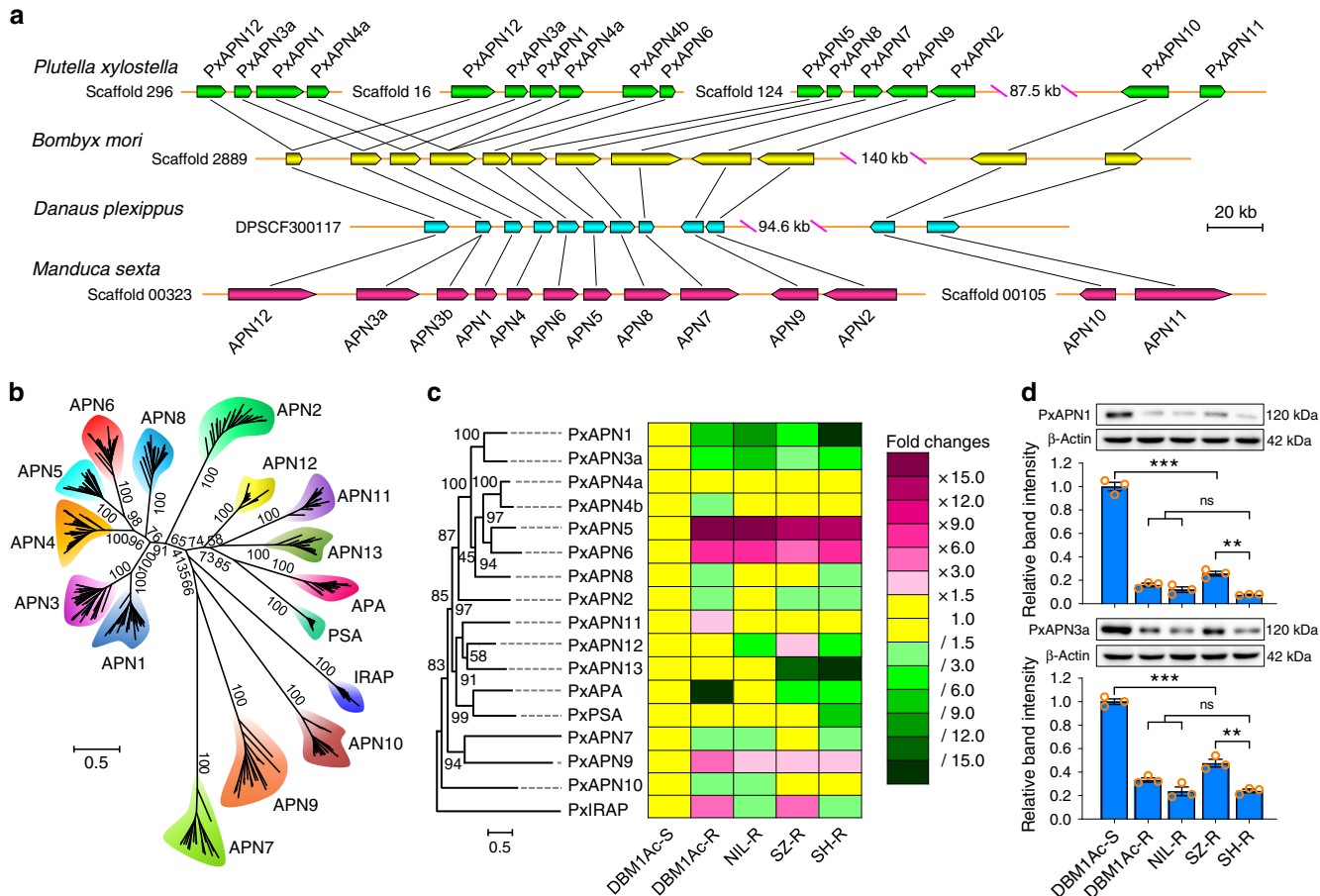

**Fig. 1 Genome-wide cloning and characterization of the APN gene family in *P. xylostella*. a** Synteny analysis of APN1-12 genes among four lepidopteran insects. **b** Phylogenetic analysis of the currently available 340 lepidopteran APN and other M1 aminopeptidases by maximum likelihood method based on the optimized LG+G model at 495 aligned amino acid positions. **c** The constitutive transcription profiles of PxAPN and other M1 aminopeptidase genes in midgut tissues of fourth-instar larvae from all the Bt-susceptible and -resistant *P. xylostella* strains as determined by qPCR analysis. For each gene, the expression fold changes are color-coded according to the gradient, magenta and green rectangles indicate significant up- and down-regulation, respectively (ratio >1.5-fold in either direction), whereas yellow rectangles indicate no significant transcription variations. Genes are organized according to their phylogenetic tree constructed by the maximum likelihood method based on the optimized LG+G+I model at 690 aligned amino acid positions. **d** The relative expression levels of PxAPN1 and PxAPN3a proteins in BBMV samples of fourth-instar larvae from different strains. Both the detection of PxAPN protein levels by Western blots (upper row) and quantitative estimation of band intensity by densitometry (graph) are presented. Data are presented as mean values (**c**) and mean values ± SEM (**d**), $n = 3$ biologically independent samples, $*p < 0.05$, $**p < 0.01$, $***p < 0.001$, ns, not significant, one-way ANOVA with Holm–Sidak's test was used in (**c**) and (**d**) for comparison. Source data are provided as a Source Data file.

**PxAPN1 and PxAPN3a are functional midgut Cry1Ac receptors**. In order to ascertain the role, if any, of the differentially expressed PxAPN genes in the mechanism of action of Bt toxins, the *PxAPN1*, *PxAPN3a*, *PxAPN5*, and *PxAPN6* genes were transiently expressed in Sf9 cells. These insect-derived cells are not naturally susceptible to the Bt Cry1Ac toxin and as a result have been used to test putative toxin receptors[43]. Specific APN activity assays showed that high levels of APN activity were detected in all the PxAPN-expressing Sf9 cells, indicating that these recombinant PxAPN proteins were biologically active (Supplementary Fig. 3a). The successful ectopic expression of these GFP-fused PxAPN proteins was further verified by Western blots (Supplementary Fig. 3b). Immunofluorescent localization detection illustrated that all four PxAPN proteins were located on the cell surface, but that only PxAPN1 and PxAPN3a proteins could bind Cry1Ac toxin (Fig. 2a). Furthermore, Cry1Ac toxin induced a time and concentration-dependent cell cytotoxicity in cells expressing PxAPN1 or PxAPN3a but not in those expressing PxAPN5 or PxAPN6 (Fig. 2b and Supplementary

Fig. 3c). These observations led to the conclusion that PxAPN1 and PxAPN3a are functional receptors of the Bt Cry1Ac toxin in *P. xylostella*.

**PxAPN1 and PxAPN3a knockdown/out confers Cry1Ac resistance.** Having established that PxAPN1 and PxAPN3a could act as receptors for Cry1Ac, we then intended to establish whether their loss could induce resistance to the toxin. RNAi-induced knockdown of *PxAPN1*, *PxAPN3a*, *PxAPN5*, and *PxAPN6* gene expression in the Bt-susceptible strain of *P. xylostella* was undertaken. The transcript levels for these four PxAPN genes were significantly reduced at 24 h post-RNAi, with the lowest expression at 48 h and lasting for about 72 h in both cases. The observed silencing was specific to each targeted gene (Fig. 2c and Supplementary Fig. 4a). Subsequent bioassays conducted at 48 h post-RNAi for 72 h revealed a remarkable reduction in the susceptibility to Cry1Ac protoxin at both 1 mg/L ($LC_{50}$ value) and 2 mg/L ($LC_{90}$ value) in both dsPxAPN1- and dsPxAPN3a-treated

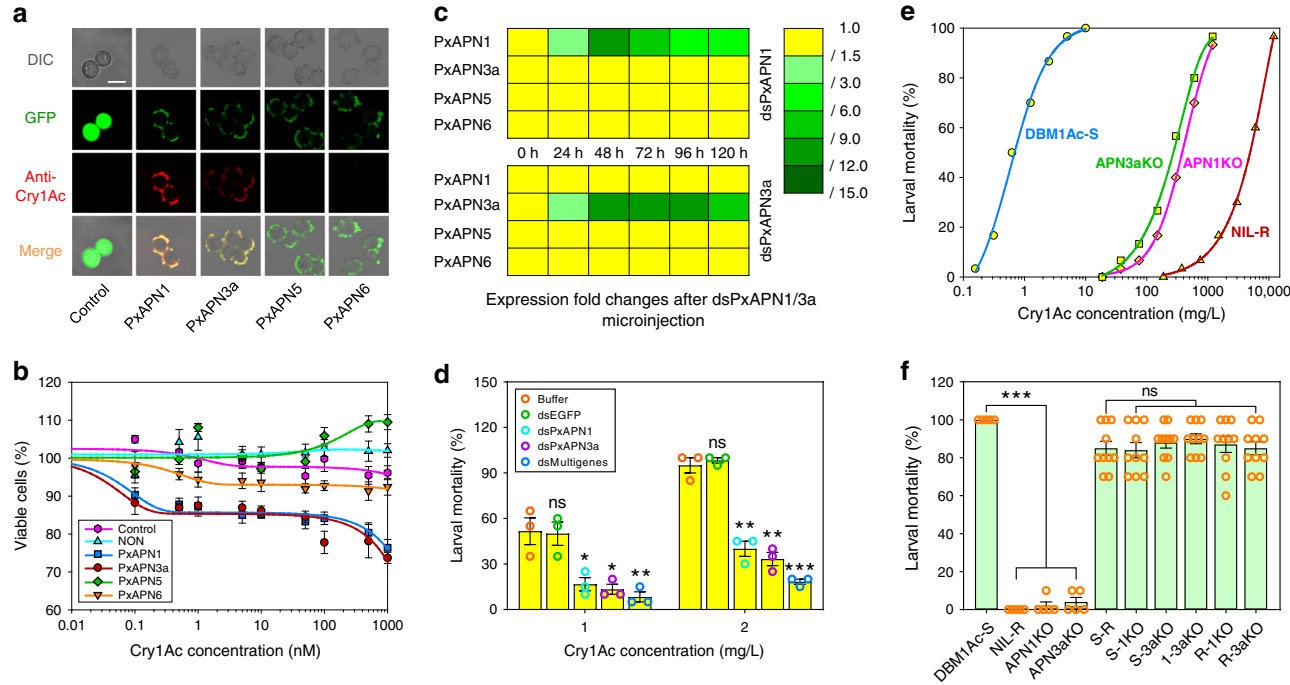

**Fig. 2 PxAPN1 and PxAPN3a are functional Cry1Ac receptors in *P. xylostella*. a** Detection of Cry1Ac binding to Sf9 cells expressing PxAPN proteins by immunolocalization. Gray panels: differential interference contrast (DIC) microscopy views; green panels: GFP-expression fluorescent signal; red panels: Cry1Ac-binding fluorescent signal. Mixed color panels: merged images from both the green and red fluorescent channels. The scale bar is 20 μm. **b** Susceptibility to a series of concentrations of Cry1Ac toxin (0.1 nM to 1 μM) in untransfected Sf9 cells (control), Sf9 cells transfected with only an empty expression vector (NON) and PxAPN-expressing Sf9 cells after 24 h incubation. **c** RNAi-mediated silencing of *PxAPN1* or *PxAPN3a* is expressed relative to transcript level for each gene at time 0 h, which are displayed as expression fold changes and color-coded according to the gradient. **d** Susceptibility to Cry1Ac protoxin in DBM1Ac-S larvae injected with buffer or dsEGFP, dsPxAPN1, dsPxAPN3a or a multiple gene mixture (dsPxAPN1 and dsPxAPN3a). **e** Non-linear log dose–response curves for *P. xylostella* larvae from the susceptible DBM1Ac-S strain, the CRISPR/Cas9-mediated *PxAPN1* and *PxAPN3a* knockout strains PxAPN1KO and PxAPN3aKO, and the near-isogenic resistant strain NIL-R exposed to Cry1Ac protoxin. **f** Interstrain allelic complementation tests with diagnostic doses of Cry1Ac protoxin (10 mg/L). F1 progeny were produced by crossing one susceptible and three resistant strains in all pair-wise combinations. Data are presented as mean values (**c**) and mean values ± SEM (**b**, **d**, **f**), $n = 6$ (**b**, **c**), $n = 3$ (**d**), and $n = 5/10$ (**f**) biologically independent samples, *$p < 0.05$, **$p < 0.01$, ***$p < 0.001$, ns, not significant, one-way ANOVA with Holm–Sidak's test was used in (**c**), (**d**), and (**f**) for comparison. Images in (**a**) are representative micrographs of three independent experiments that produced similar results. Source data are provided as a Source Data file.

groups compared to the controls. Furthermore, the use of combinational RNAi to silence both PxAPN genes simultaneously by microinjection of a combination of dsPxAPN1 and dsPxAPN3a (dsMultigenes) caused incremental increases in Cry1Ac tolerance at both doses (Fig. 2d). In contrast, no susceptibility changes to Cry1Ac protoxin were detected in dsPxAPN5- and/or dsPxAPN6-treated groups compared to the controls (Supplementary Fig. 4b).

CRISPR/Cas9-mediated knockout of both *PxAPN1* and *PxAPN3a* genes was further utilized in vivo to validate their roles in Cry1Ac resistance. We used an optimized germline transformation and mutation screening strategy to successfully construct stable homozygous mutant strains of *PxAPN1* and *PxAPN3a* genes designated APN1KO and APN3aKO, respectively (Supplementary Table 4 and Supplementary Fig. 5). Subsequent bioassays showed that APN1KO and APN3aKO larvae exhibited approximately 463-fold and 346-fold levels of resistance to Cry1Ac protoxin, respectively (Fig. 2e), confirming that the disruption of either PxAPN1 or PxAPN3a can confer high-level Cry1Ac resistance in *P. xylostella*. Genetic complementation tests between the knockout strains and the NIL-R strain found that F1 progeny from mass crosses between pairs of strains had high larval mortality at the discriminating concentration (10 mg/L) of Cry1Ac protoxin. This indicated that resistance

was effectively recessive for each allele but also that resistance in NIL-R was not due to a mutation in PxAPN1 or PxAPN3a (Fig. 2f).

**Reduced *PxAPN1* and *APN3a* expression is linked to resistance.** Genetic linkage analysis was conducted to test the cosegregation of differential expression of *PxAPN1*, *PxAPN3a*, *PxAPN5*, and *PxAPN6* genes with Cry1Ac resistance in the NIL-R strain. qPCR analysis showed that the transcript levels of *PxAPN1* and *PxAPN3a* genes in both Cry1Ac-unselected F2 backcross families exhibited two distinct groups: one group displayed reduced expression of *PxAPN1* and *PxAPN3a* genes, whereas the other group showed transcript levels resembling larvae from the susceptible parental strain or the F1 progeny (Fig. 3a, b). Moreover, the ratio between the numbers of individuals in each group was 9:9 in both backcross families for the *PxAPN1* and *PxAPN3a* genes, which was a perfect fit to the theoretically random assortment ratio 1:1 ($p = 1.0$; $\chi^2$ test). In stark contrast, all of the survivors from Cry1Ac-treated F2 backcross families had reduced *PxAPN1* and *PxAPN3a* transcript levels compared to larvae from the DBM1Ac-S strain or the F1 progeny, indicating tight linkage (cosegregation) with Cry1Ac resistance in NIL-R ($p < 0.001$, $\chi^2$ test). The transcript levels for *PxAPN5* and *PxAPN6* genes in both Cry1Ac-treated and non-treated groups remained significantly

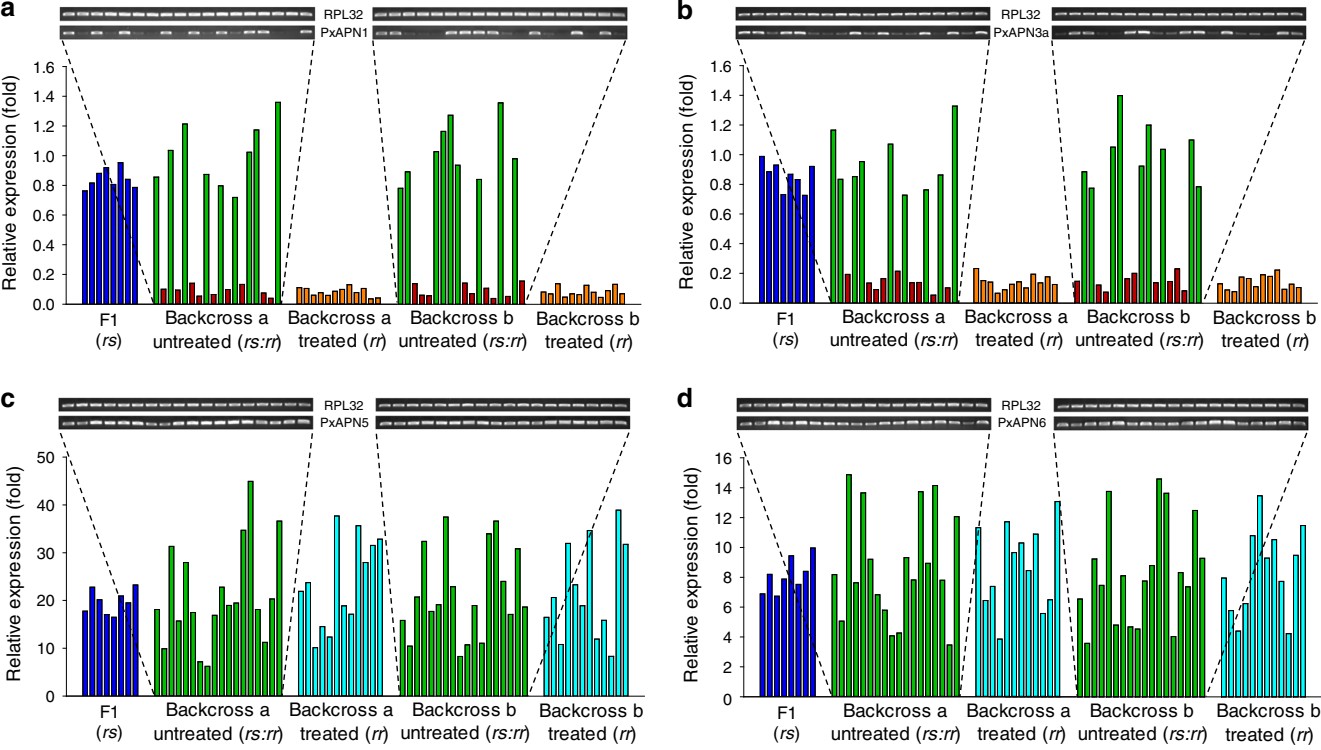

**Fig. 3 Linkage of Cry1Ac resistance phenotype to reduced *PxAPN1* and *PxAPN3a* expression.** Expression levels for the F1, Cry1Ac-selected and non-selected backcross families are relative to levels in the susceptible DBM1Ac-S strain. Corresponding intensity of amplified transcript bands for the *PxAPN1* (**a**), *PxAPN3a* (**b**), *PxAPN5* (**c**), *PxAPN6* (**d**) and the internal standard *RPL32* gene are shown above each graph. Both results using larvae of the subfamily generated from backcrossing a female from F1 and a male from the NIL-R strain (backcross family a) and using larvae of the subfamily generated from backcrossing a male from F1 and a female from the NIL-R strain (backcross family b) are shown. Source data are provided as a Source Data file.

up-regulated and did not associate with Cry1Ac resistance (Fig. 3c, d).

**MAPK-induced differential expression of diverse midgut genes.** We previously established that an activated MAPK cascade *trans*-regulates differential expression of *PxmALP*, *PxABCC1*, *PxABCC2*, *PxABCC3*, and possibly other midgut genes involved in Cry1Ac resistance in *P. xylostella*[26–28]. Hence, silencing of *PxMAP4K4* expression in resistant NIL-R larvae was performed to ascertain whether this MAPK-mediated mechanism also accounts for the differential expression of PxAPN and other midgut genes to cause Cry1Ac resistance. After dsRNA injection in the Bt-resistant strain, the transcript levels of *PxMAP4K4* were substantially reduced at 48 h post-RNAi (Fig. 4a). Correspondingly, the transcript levels of *PxmALP*, *PxABCB1*, *PxABCC2*, *PxABCC3*, *PxABCG1*, *PxAPN1*, and *PxAPN3a* were increased, while the transcript levels of *PxABCC1*, *PxAPN5*, and *PxAPN6* decreased at 48 h post-injection (Fig. 4a). Bioassays conducted at 48 h post-RNAi for 72 h indicated that silencing of *PxMAP4K4* expression led to significantly increased larval susceptibility to Cry1Ac protoxin compared to the controls (Fig. 4b). Therefore, *PxMAP4K4* gene silencing recovered the expression of PxAPN1, PxAPN3a and other midgut proteins and restored Cry1Ac susceptibility in NIL-R larvae.

**Increased titers of insect hormones in resistant strains.** Previous work had shown that exogenous hormone treatments can alter the transcript levels of two isoforms of *AjAPN1* genes in *Achaea janata*[41], so we decided to investigate whether insect hormones might be the upstream activation signals of the MAPK

cascade to *trans*-regulate differential expression of PxAPN and other midgut proteins thereby mediating Cry1Ac resistance in *P. xylostella*. We initially quantified hormone titers in *P. xylostella* by optimized multiple reaction monitoring (MRM)-based ultra-performance liquid chromatography–tandem mass spectrometry (UPLC–MS/MS). Hormone detection showed specific single peaks for JHs, 20E and their respective analogs methoprene and 22S, 23S-homobrassinolide in the MRM mass chromatograms (Fig. 5a). As expected, the hormone titers fluctuated during different developmental stages with clear JH II peaks observed around larval molting, and the sensitivity of this initial assay only resulted in changes in 20E levels being observed during pupation (Fig. 5b). A more detailed analysis of hormone titers during the third- and fourth-instar stages showed sharp drops in 20E and small peaks in JH II during the molting stage, in contrast, the levels of both hormones did not fluctuate significantly during the feeding intermolt stage (Fig. 5c, d). When the titers of JH II and 20E in feeding third/fourth stage larvae from the resistant and susceptible strains were compared, it was found that the levels were higher in the NIL-R strain than the DBM1Ac-S strain (Fig. 5c, d). Furthermore, it was found that they were also higher in fourth-instar feeding larvae of all the resistant strains, which correlated well with their resistance ratios (Fig. 5e, f), and hinted at the potential involvement of insect hormones in Cry1Ac resistance in *P. xylostella*.

**The MAPK cascade is modulated by insect hormonal crosstalk.** To investigate a potential causal link between hormone titer and resistance, exogenous JH analog (JHA) methoprene (200 ng), 20E (2 ng), or a mixture of the two, were microinjected into newly

molted third-instar DBM1Ac-S larvae. qPCR analysis showed that the expression of *PxMAP4K4* was significantly decreased at 48 h after methoprene treatment, whereas it had increased at 48 h after 20E treatment (Fig. 6a). The transcript levels of four PxAPN and other midgut genes were increased at 48 h after methoprene treatments, whereas their levels were notably decreased at 48 h

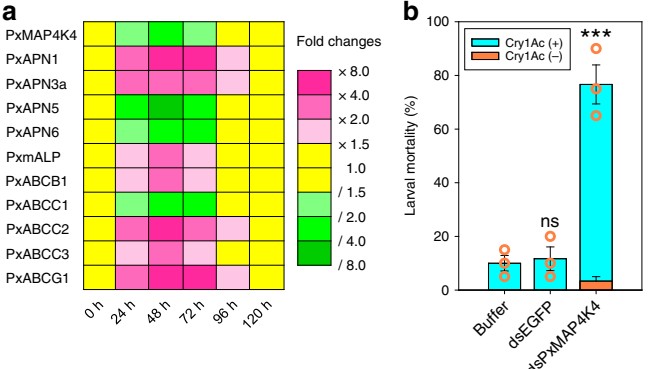

**Fig. 4 Effect of *PxMAP4K4* gene silencing on midgut gene expression and larval Cry1Ac susceptibility. a** The effect of *PxMAP4K4* gene silencing on expression of midgut genes at different periods. Gene expression level at 0 h post-RNAi was assigned a value of 1 for comparison, which are displayed as expression fold changes and color-coded according to the gradient. **b** Susceptibility to Cry1Ac protoxin (LC$_{10}$, 1000 mg/L) in resistant NIL-R larvae post-RNAi. Data are presented as mean values (**a**) and mean values ± SEM (**b**), $n = 3$ biologically independent samples, ***$p < 0.001$, ns, not significant, one-way ANOVA with Holm–Sidak's test was used for comparison. Source data are provided as a Source Data file.

after 20E treatments. However, when larvae were synchronously treated with both methoprene and 20E, while the transcript levels of *PxMAP4K4*, *PxAPN5*, *PxAPN6*, and *PxABCC1* were increased at 48 h post-microinjection, those of *PxAPN1*, *PxAPN3a*, *PxmALP*, *PxABCB1*, *PxABCC2*, *PxABCC3*, and *PxABCG1* had decreased (Fig. 6a). Intriguingly, this pattern of differential expression exactly, but inversely, mimicked that seen in the NIL-R strain following *PxMAP4K4* knockdown (Fig. 4a). To investigate whether downstream elements of the MAP4K cascade (notably p38, ERK, and JNK) might be involved in the control of expression of the midgut proteins, a combination of methoprene, 20E, and a MAPK inhibitor cocktail targeting the above three kinases was microinjected into DBM1Ac-S larvae. Despite the expected increase in expression of *PxMAP4K4* following this treatment, the expression levels of all of the tested midgut genes either remained the same as the control or only showed modest changes (Fig. 6a). To confirm the downstream effects of hormone treatment on the MAPK cascade, Western blots were performed to measure the levels of total and phosphorylated p38, ERK, and JNK (Fig. 6b). These data showed a significant increase in phosphorylated protein following treatment with 20E or a combination of 20E and methoprene. These increases were not observed when the MAPK inhibitor cocktail was included alongside the hormone mixture, results consistent with the qPCR data in suggesting that the hormones are exerting their effect via the MAP4K cascade. Bioassays performed at 48 h post-microinjection demonstrated that methoprene treatment resulted in significantly increased larval susceptibility to Cry1Ac protoxin compared to the controls, in contrast, 20E treatment caused significantly reduced susceptibility (Fig. 6c). Treatment with both methoprene and 20E simultaneously led to a more greatly reduced susceptibility to Cry1Ac protoxin than 20E

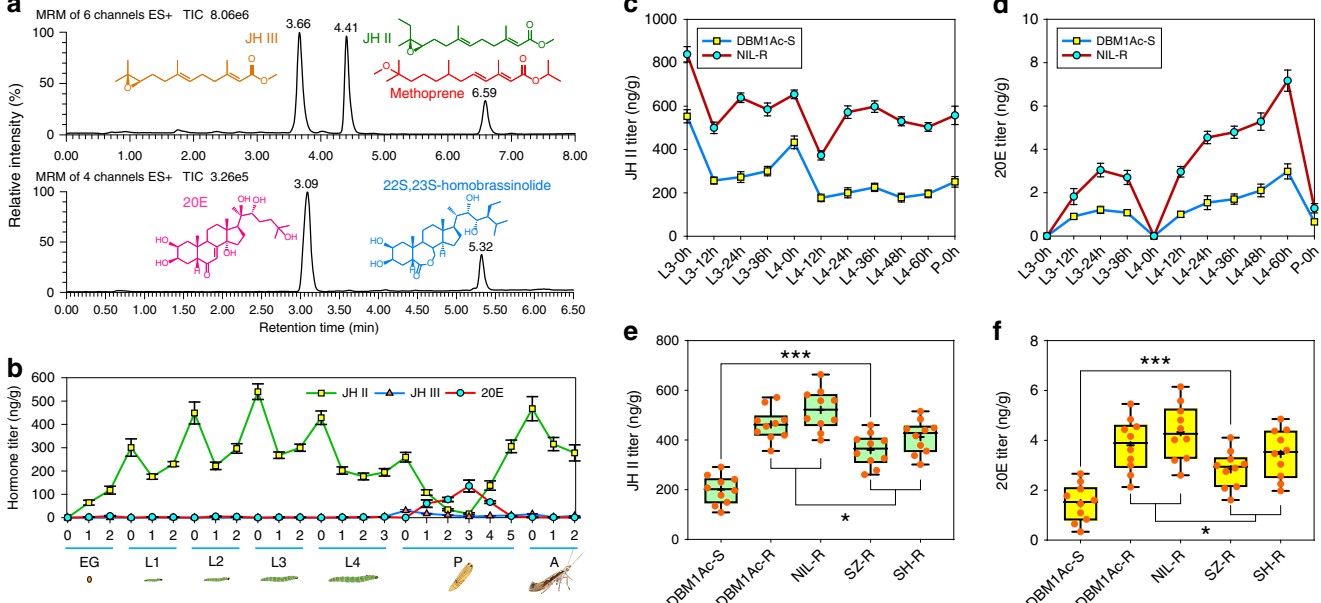

**Fig. 5 Detection of hormone titers in susceptible and resistant *P. xylostella*. a** Representative MRM chromatograms of the standard solutions of JHs (JH II and JH III, 5 ng), 20E (2 ng), methoprene (0.1 ng), and 22S, 23S-homobrassinolide (0.1 ng) detected by UPLC–MS/MS. **b** Hormone titers in whole developmental stages of the DBM1Ac-S strain. EG: eggs; L1-4: first- to fourth-instar larvae; P: pupae; A: adults. Samples were collected daily in each stage for hormone titer detection. JH II (**c**) and 20E (**d**) titers from third- to fourth-instar larvae of both DBM1Ac-S and NIL-R strains. Samples were collected every 12 h in both stages for accurate detection of hormone titer changes. **e**, **f** Box plots show JH II (**e**) and 20E (**f**) titers in all of the susceptible and resistant *P. xylostella* strains. Each orange dot represents mean hormone titer in two 1-day-old fourth-instar larvae and the black plus sign indicates mean hormone titer of all the detected individuals in each strain. Median and quartile values are provided by the central line and box boundaries, whiskers show min to max values. Data are presented as mean values ± SEM, $n = 3$ (**b**–**d**) and $n = 10$ (**e**, **f**) biologically independent samples, *$p < 0.05$, ***$p < 0.001$, one-way ANOVA with Holm–Sidak's test was used in (**e**) and (**f**) for comparison. Source data are provided as a Source Data file.

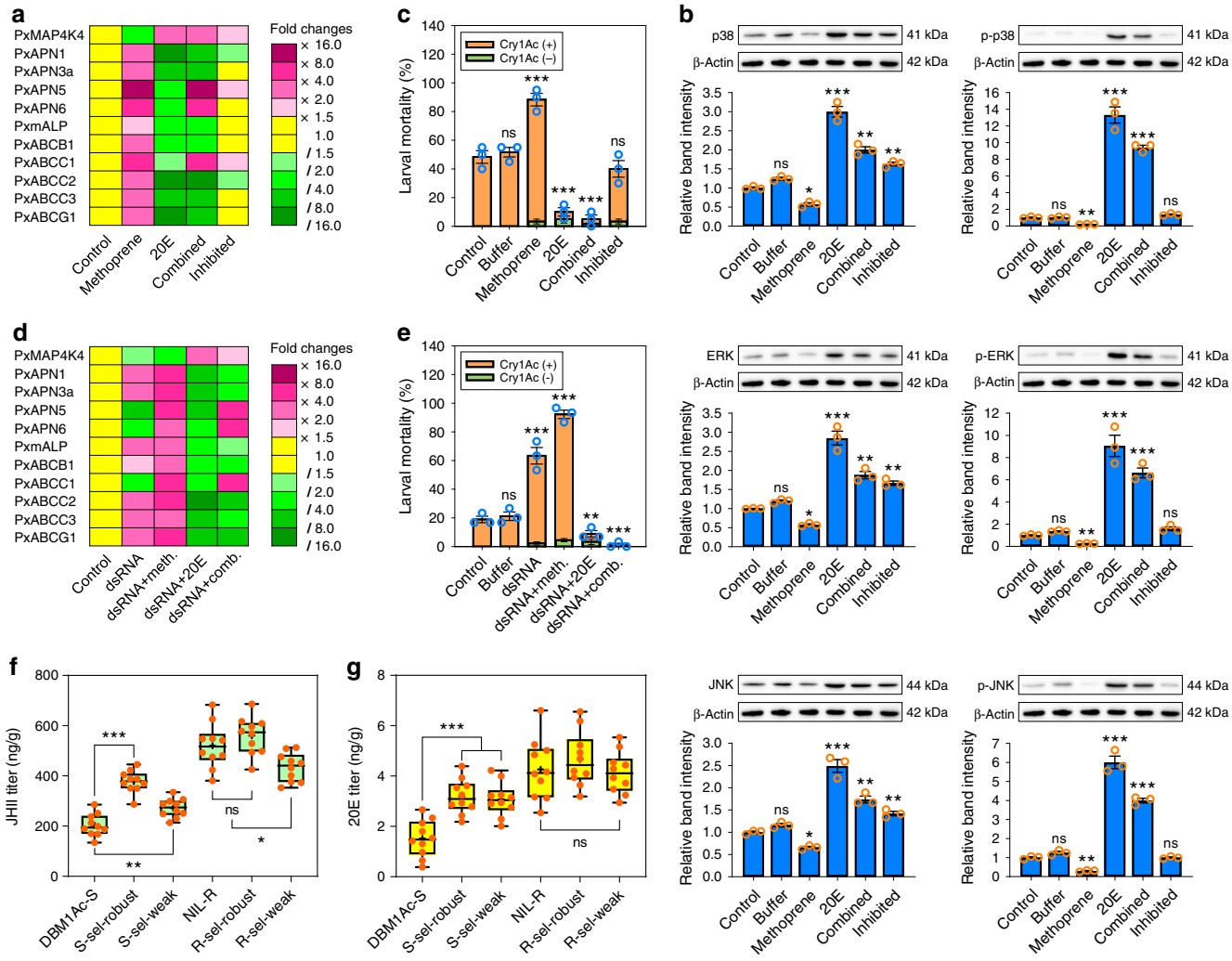

**Fig. 6 Effect of hormone titers in susceptible and resistant *P. xylostella*. a** Effect of methoprene and 20E treatments on gene expression levels in the injected DBM1Ac-S larval midguts. Gene expression level at 0 h was assigned a value of 1 for comparison, which are displayed as expression fold changes and color-coded according to the gradient. **b** Effect of methoprene and 20E treatments on protein expression and phosphorylation levels of three key MAPK downstream kinases p38, JNK, and ERK in the injected DBM1Ac-S larval midguts. Both the detection of protein and phosphorylation levels by Western blots (upper row) and quantitative estimation of band intensity by densitometry (graph) are presented. **c** Susceptibility to Cry1Ac protoxin ($LC_{50}$, 1 mg/L) in the exogenous hormone-treated DBM1Ac-S larvae. **d, e** Effect of exogenous hormone treatment on midgut gene expression levels (**d**) and susceptibility to Cry1Ac protoxin ($LC_{20}$, 0.3 mg/L) (**e**) in the *PxMAP4K4*-silenced DBM1Ac-S larvae. **f, g** Box plots show the detection of Cry1Ac toxin-induced JH II (**f**) and 20E (**g**) titers in unselected or Cry1Ac-selected robust and weak DBM1Ac-S and NIL-R larvae. Each orange dot represents mean hormone titer in two 1-day-old fourth-instar larvae and black plus sign indicates mean hormone titer of all the detected individuals in each strain. Median and quartile values are provided by the central line and box boundaries, whiskers show min to max values. Data are presented as mean values (**a**, **d**) and mean values ± SEM (**b**, **c**, **e**–**g**), n = 3 (**a**–**e**) and n = 10 (**f**, **g**) biologically independent samples, *p < 0.05, **p < 0.01, ***p < 0.001, ns, not significant, one-way ANOVA with Holm–Sidak's test was used for comparison. Source data are provided as a Source Data file.

treatment alone, and this effect was negated by simultaneous treatment with the MAPK inhibitor cocktail (Fig. 6c).

To further probe the interplay between hormone exposure and the MAPK cascade, we studied the effect of hormone treatment on insects in which *PxMAP4K4* expression had been reduced by RNAi. In particular, we were interested in whether 20E induced loss of susceptibility could be reversed by silencing of *PxMAP4K4*. Figure 6d, e shows that the effect of 20E on the silenced strain was similar to the effect on the non-silenced strain. A similar observation was made with the dual hormone treatment. Given that the data presented in Fig. 6a, c clearly indicate an association between hormone treatment and *PxMAP4K4* expression, we surmise that the 20E-induced increase in *PxMAP4K4* transcription can overcome the effect of RNAi-induced reduction in transcript levels.

**Associations among hormone titer, Bt resistance, and fitness.** Based on the above observation that exposure to methoprene and 20E could directly affect resistance levels to Bt, and induce the same changes in midgut proteins as seen in the resistant strains, we proposed a model. In this model, elevated levels of 20E can induce resistance to Bt through blocking the expression of many midgut proteins, in particular, those which act as functional Bt toxin receptors. Fitness costs related to life-history traits frequently occur during insect resistance to Bt infection[44], therefore, resistance evolution without growth penalty is extremely important for successfully overcoming Bt pathogenesis. Thus, we suggest that fitness costs associated with this 20E-induced physiological change are modulated by a corresponding increase in JH which results in the restoration of midgut physiology through the expression of non-receptor paralogs. To test this

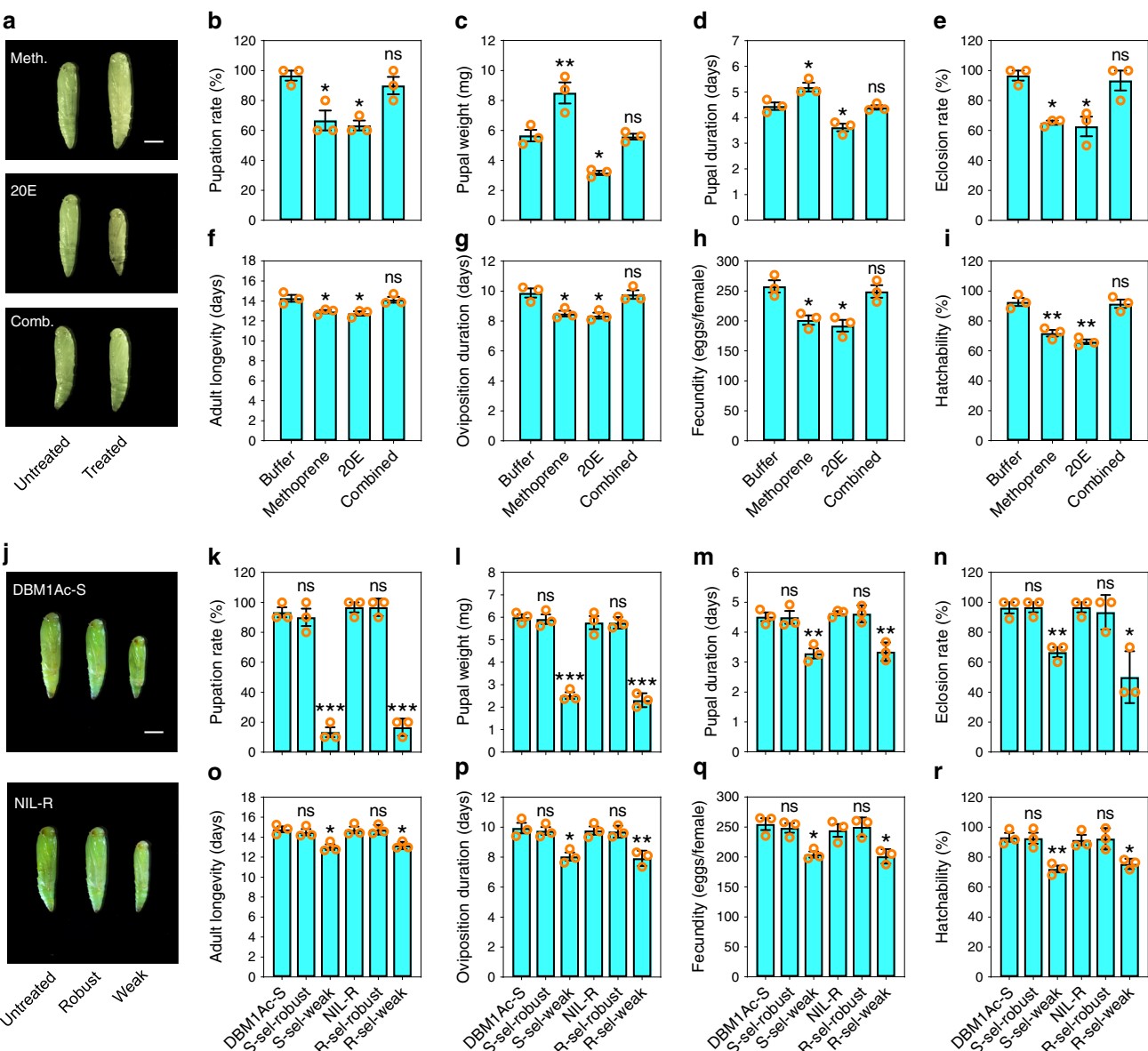

**Fig. 7 The effect of hormone or Cry1Ac exposure on life-history traits in *P. xylostella*. a** Representative pupal morphology after exogenous treatment (on day 0 of the third-instar larvae) with methoprene (upper box), 20E (middle box) or both (lower box). The scale bar is 1 mm. **b–i** Analysis of fitness costs of exogenous hormone treatments. **j** Pupal morphology after exogenous treatment of third-instar larvae with an LC$_{50}$ dose of Cry1Ac protoxin for DBM1Ac-S (upper box) and NIL-R (lower box). The scale bar is 1 mm. **k–r** Analysis of fitness costs of exogenous Cry1Ac treatments. Data are presented as mean values ± SEM, $n = 3$ biologically independent samples, $*p < 0.05$, $**p < 0.01$, $***p < 0.001$, ns, not significant, one-way ANOVA with Holm–Sidak's test was used for comparison. Source data are provided as a Source Data file.

model, we exposed larvae to methoprene and 20E either individually or in combination and measured various life-history traits (Fig. 7a–i). These data support our model in that exposure to the individual hormones resulted in significantly reduced pupation rates, eclosion rates, adult longevity, oviposition time, female fecundity, and egg hatchability, whereas those in the hormone combination were indistinguishable from the buffer control (Fig. 7b, e–i). Methoprene resulted in larger pupae, but which stayed in that state for longer, whereas 20E gave smaller pupae and a shorter time in this stage, once again, the combination treatment was indistinguishable from the control (Fig. 7a, c, d). Moreover, we observed that the 20E-treated third-instar larvae molted early to the fourth-instar and pupal stages with reduced appetite and food intake, while methoprene-treated third-instar larvae behaved completely opposite, i.e., molted late to the fourth-

instar and pupal stages with increased appetite and food intake. In contrast, the development time and feeding behavior of third-instar larvae treated with both hormones were comparable to those in the control group. Thus, these differences might attribute to the observed variation in the resultant pupal size (Fig. 7a). Since it is known that exposure to Bt can induce tolerance to Bt[45], and that toxin can induce the expression of *PxMAP4K4* gene in the DBM1Ac-S strain[26], we further examined the effect of Bt toxin exposure on hormone levels and fitness. Both newly molted third-instar DBM1Ac-S and NIL-R larvae were exposed to LC$_{50}$ levels of Cry1Ac protoxin for 72 h. Surviving fourth-instar larvae showed two quite distinct groups in larval weight, some were significantly lighter than the others and were termed "weak", whereas the others were indistinguishable from untreated larvae and were termed "robust". For the robust larvae, it was found that

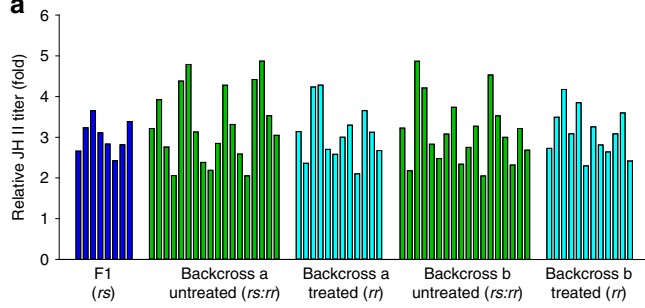

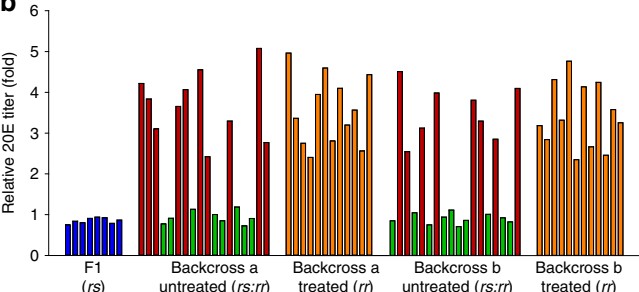

**Fig. 8 Linkage of Cry1Ac resistance phenotype to increased 20E titers in the NIL-R strain.** JH II (**a**) and 20E (**b**) titers for the F1, Cry1Ac-selected and non-selected backcross families are relative to hormone titers in the susceptible DBM1Ac-S strain. Both results using larvae of the subfamily generated from backcrossing a female from F1 and a male from the NIL-R strain (backcross family a) and using larvae of the subfamily generated from backcrossing a male from F1 and a female from the NIL-R strain (backcross family b) are shown. Source data are provided as a Source Data file.

while exposure of the DBM1Ac-S larvae resulted in significant increases in the levels of both hormones, there were no significant increases seen in the NIL-R larvae (Fig. 6f, g). For the weak larvae, while an increase in 20E in DBM1Ac-S was observed that was comparable to that seen in the robust larvae, the increase in JH II was significantly less. The levels of 20E in weak NIL-R specimens were equivalent to the robust and untreated control, however, a lower level of JH II was found (Fig. 6f, g). As expected, when the life-history traits of the surviving insects were measured, it was found that the robust individuals were indistinguishable from the untreated controls, whereas the weak ones were significantly affected in all the tested parameters (Fig. 7j–r).

**Linkage of increased 20E titers with Cry1Ac resistance**. The data presented above suggest a causal association between increased titers of insect hormones and receptor-mediated resistance to Bt Cry toxin, a link which involves the MAPK signaling cascade. To further strengthen this association, we used genetic linkage analysis to determine co-segregation of increased titers of JH II and 20E with Cry1Ac resistance in the NIL-R strain. UPLC–MS/MS detection showed that JH II titers in both Cry1Ac-treated and non-treated backcross groups had the same level of up-regulation as the resistant parental strain, indicating a lack of association with Cry1Ac resistance (Fig. 8a). In stark contrast, 20E titers in both Cry1Ac-unselected F2 backcross families exhibited two distinct groups: one group displayed increased 20E titers, whereas the other group showed 20E levels resembling larvae from the susceptible parental DBM1Ac-S strain or the F1 progeny (Fig. 8b). Moreover, the ratio between the numbers of individuals in each group, 10:8 in backcross family a and 8:10 in backcross family b, followed the 1:1 random assortment ratio ($p > 0.10$; $\chi^2$ test). Crucially, all of the survivors from the Cry1Ac-

treated F2 backcross families had increased 20E titers, indicating tight linkage (cosegregation) with Cry1Ac resistance in NIL-R ($p < 0.001$, $\chi^2$ test).

## Discussion

The proteinaceous Cry toxins produced by Bt represent highly potent virulence factors which when presented to an insect at a high dose, such as in Bt crops, overwhelm the insect's immune system leading to death[3]. Through modification of the toxin binding protein (receptor) on the surface of its midgut epithelium, an insect can effectively block the action of the toxin leading to high levels of resistance[19]. The first Cry toxin receptor identified in an insect was an APN protein in *Manduca sexta*[20,30]. In this work, we show that two isoforms of this enzyme can act as functional receptors in *P. xylostella* and that reduction in their expression leads to loss of susceptibility towards Cry1Ac. We have also conducted the genome-wide characterization of the APN gene family and unified their nomenclature and classification in insects. Interestingly, we discovered a syntenic APN gene cluster, containing 12 or more APN genes in different lepidopteran insects, is estimated to have been present since the Jurassic period about 150 million years ago[46]. Our previous studies have shown that silencing of other potential *P. xylostella* Cry toxin receptor genes including *PxmALP*, *PxABCB1*, *PxABCC2*, *PxABCC3*, and *PxABCG1* also results in decreased Cry1Ac susceptibility[26–28].

One model for Cry toxin mechanism of action involves sequential binding of the toxin to different receptors[19], whereas others suggest that multiple receptors in a susceptible host represent alternative receptors rather than members of a sequential pathway[47]. In a sequential model, one would predict that blocking any of the steps would lead to a similar level of resistance. In this work, we show that knocking out either APN receptor, whilst leading to high levels of resistance, does not match the level of resistance seen in the parent strain, suggesting at least some degree of receptor redundancy. The high level of resistance in the parent strain can be associated with the simultaneous down-regulation of multiple Cry toxin receptors. Whilst the loss of physiologically important proteins might be expected to impart a fitness cost to the insect, it is noteworthy that the down-regulation of proteins known to act as toxin receptors is accompanied by the up-regulation of paralogs that have no receptor function. Gene duplication and subsequent transcriptional plasticity is a major driving force for adaptive evolution of insect response to environmental stress[48,49]. Specifically, up-regulation of *PxAPN5* and *PxAPN6* might compensate for the down-regulation of *PxAPN1* and *PxAPN3a* to diminish the fitness costs of Cry1Ac resistance and perhaps resembling the differential alteration of *TnAPN1* and *TnAPN6* in resistant *Trichoplusia ni*[50]. It is noteworthy that the total midgut APN enzyme activities were similar in our susceptible and resistant strains[26] and we did not detect any fitness costs in our near-isogenic Cry1Ac-resistant NIL-R strain[51]. Previously, we found that a MAPK signaling pathway can *trans*-regulate the differential expression of *PxmALP*, *PxABCC1-3*, and possibly *PxABCB1* and *PxABCG1* genes thereby resulting in Cry1Ac resistance in four different *P. xylostella* strains[26–28]. We have now identified that this pathway can also *trans*-regulate the differential expression of four PxAPN genes to mediate Cry1Ac resistance in the same four resistant *P. xylostella* strains.

Although the mechanism underlying the altered MAPK signaling pathway is unknown, in this work, we discovered an intriguing causal link to the insect hormones 20E and JH. Levels of both insect hormones were significantly increased in all of our resistant strains of *P. xylostella*, but more relevantly when

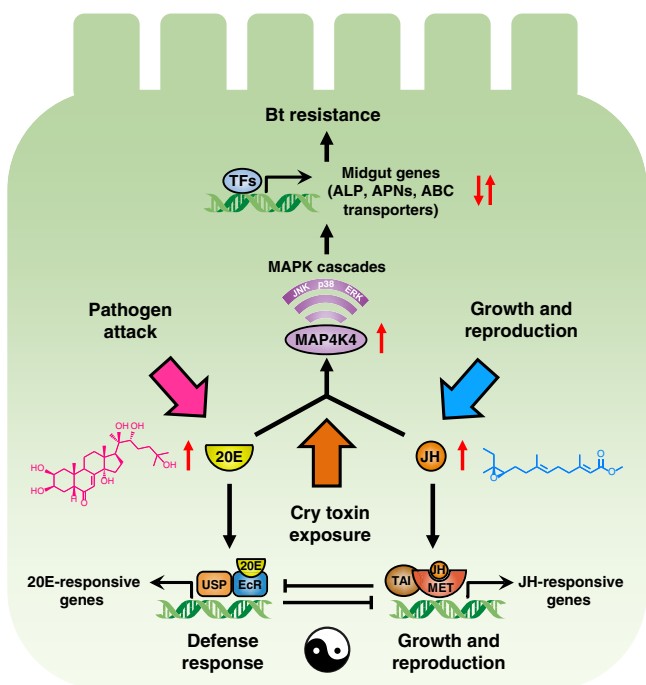

**Fig. 9 Model for insect hormones-mediated growth-defense tradeoffs orchestrating Cry1Ac resistance in *P. xylostella*.** The molecular basis of JH and 20E signaling pathways is highly conserved in different insects: they have antagonistic actions to control life-history traits including growth, development, reproduction, and immunity. Upon exposure of *P. xylostella* to Bt Cry1Ac toxin, in the midgut epithelial cells, we can speculate that JH/20E can bind to the respective heterodimeric receptor complex composed of Methoprene-tolerant (Met) and its partner Taiman (Tai) or ecdysone receptor (EcR) and Ultraspiracle (USP) to initiate the transcription of JH/20E-responsive genes. Meanwhile, increased levels of JH and 20E can activate the MAPK signaling cascades to *trans*-regulate differential expression of multiple midgut genes including four PxAPNs (*PxAPN1, PxAPN3a, PxAPN5*, and *PxAPN6*), *PxmALP, PxABCB1, PxABCC1-3*, and *PxABCG1* via manipulating unidentified transcription factors (TFs)[70], which finally results in high-level Bt resistance without growth penalty in *P. xylostella*.

susceptible insects were exposed to both hormones: differences in gene expression of *PxMAP4K4, PxAPN1,3a,5,6, PxABCB1, PxABCC1,2,3, PxABCG1* and *PxmALP* all mirrored those seen in the resistant strain, and a decrease in toxin susceptibility was observed. The MAPK inhibitor assays indicated that insect hormones can orchestrate the expression of these resistance-related midgut genes via the MAPK signaling cascade, which strengthens the case for a causal link between insect hormones and MAPK signaling cascade. We confirmed that increased 20E, but not JH II, titers were genetically linked to Cry1Ac resistance in *P. xylostella*, which provided more direct evidence that increased titers of insect hormones in resistant *P. xylostella* strains is actually a cause rather than an effect of resistance. In this regard, we can ask: why JH II titers were also increased when only increased 20E titers were tightly linked to Cry1Ac resistance?

Besides modulating life-history traits including growth, development, and reproduction, JH and 20E can also participate in insect host immune defense and resistance to pathogenic infection: JH suppresses stress resistance and immunity, whereas 20E activates these processes[38]. The insect immune system is like a double-edged sword: on the one side, its activation can effectively fight against infections from a broad range of pathogens, but on the other, its activation has a negative impact on insect growth

and development[52]. Hence, maintaining both immune defense and life-history traits is physiologically and energetically costly, and the tradeoff between them is crucial for limiting resource allocation in diverse insects[39]. A similar phenomenon is well established in plants where the growth-defense tradeoff is a fine balance and one which is modulated by MAPK-dependent hormonal signaling pathways[53−56]. Similar tradeoffs have also been identified in vertebrates[57,58], indicating that hormonal signaling plasticity is a general cross-kingdom strategy for thwarting pathogens.

As mentioned above, an imbalance in the ratio of the two insect hormones can shift the insect's physiology towards either growth/reproduction or defense with a corresponding cost on the other process, and our data in which fitness costs were associated with individual hormone exposure support that. We are proposing a fine-tuning of this process in which a balanced increase in the two hormones affects the insect in such a way that it loses susceptibility to the Cry toxin whilst maintaining gut homeostasis through the expression of paralogous genes. We demonstrated that a balanced rise in both hormones is induced by sub-lethal exposure to Bt toxin and also found, that via some unknown genetic changes, a constitutive and balanced rise has occurred in the resistant strains. A similar case was documented in *Drosophila virilis*, in which increased titers of 20E and JH were identified as an induced adaptive response to the unfavorable environmental heat stress by decreasing egg production and oviposition. Moreover, in a heat stress mutant strain of *D. virilis*, elevated JH and 20E levels caused by an as yet uncharacterized gene mutation in chromosome 6 resulted in the same reduced fertility phenotype as observed for the wild type strain under heat stress[59,60]. In *P. xylostella*, we hypothesize that a balanced rise in the levels of the two hormones pushes the insect towards a growth-defense tradeoff instead of a full-scale defense response to successfully evolve a Bt resistance phenotype, and we present a model of this proposed system in Fig. 9. Although Bt Cry toxin has been shown to induce various aspects of an immune response[45], our previous comparisons of transcript levels between susceptible and resistant *P. xylostella* strains did not find any examples of immune-related genes such as peptidoglycan recognition proteins (PGRPs), antimicrobial peptides (AMPs), or phenoloxidases (POs) in the list of genes that had significantly increased expression in the resistant strain[32,33,61].

A previous study using Bt-resistant *Helicoverpa armigera* larvae found that there was an increased titer of JH in the resistant strain[62]. Although that study indicated that this hormone was directly associated with fitness costs in the resistant strain, it did not look at levels of 20E nor establish any causal link between JH titer and Bt-resistance. Detailed work on Bt tolerance in *Ephestia kuhniella* found that non-genetically inherited tolerance could be induced by exposure to sub-lethal concentrations of Bt and vertically transmitted through maternal effect[45]. This work had noted that while the initial elevated immune response in this insect carried significant developmental penalties, continued exposure to Bt led to a reduced developmental penalty while tolerance was maintained[63]. Our results are consistent with a model in which tolerance and resistance to Bt are linked and both associated with hormonal regulation. The mechanistic basis of the interplay between hormone levels, MAPK signaling pathway and Bt resistance remains unclear. While there is evidence for altered protein phosphorylation levels of p38, JNK, and ERK in the MAPK signaling cascade triggering the differential expression of the midgut genes, it is less clear how the hormones feed into this process. Our data here indicate that changes in JH and 20E concentrations can affect MAP4K4, p38, JNK, and ERK, which suggests that they are acting upstream of the MAPK signaling cascade. However, in the resistant strains, the causative mutation

maps to the *MAP4K4* locus and most likely directly affects the expression of this gene[26]. The fact that JH and 20E levels are also increased in these strains might suggest that their levels are tightly coordinated with MAP4K4 levels in some way. Although the MAPK signaling pathway is strongly linked to down-regulation of the Bt toxin receptors, it is quite possible that the two hormones are affecting other aspects of the insect's response via other routes. Initial exposure to Bt can trigger an immediate defense response, that may come with fitness costs, but then a fine-tuning between 20E and JH can balance that tolerance while reducing those costs. Where that balance is not maintained, e.g., in some of our larvae exposed to Bt toxins, severe fitness costs prevail. Differences between the weak and robust larvae may reflect different levels of intoxication and a more extreme, and costly, response to deal with excessive exposure. Continued exposure can result in genetic mutation that leads to constitutive expression of the defense response now manifested as resistance. In other cases of acquired resistance, the phenotype can be as a result of less subtle changes, such as functional deletions of a receptor. As suggested in a recent review[64], we should be looking to better integrate work on immunity, toxicology, and endocrinology to better understand the host–pathogen interactions between insects and Bt. Our current work linking endocrine associated growth/defense tradeoffs with Bt tolerance/resistance should greatly facilitate future research in this emerging field.

## Methods

**Insect strains**. Five *P. xylostella* strains including one susceptible strain DBM1Ac-S and four resistant strains DBM1Ac-R, NIL-R, SZ-R, and SH-R were used in this study[26,65,66]. The DBM1Ac-R, NIL-R, and SZ-R strains have developed approximately 3500-, 4000-, and 450-fold resistance to Cry1Ac protoxin compared to the susceptible DBM1Ac-S strain, while SH-R larvae have developed about 1900-fold resistance to Bt var. *kurstaki* (Btk) formulation compared to the DBM1Ac-S strain. The field-evolved or laboratory-selected resistance to Bt Cry1Ac toxin or Btk formulation in these four independent *P. xylostella* strains has a similar mechanism involving MAPK-mediated differential expression of *PxmALP*, *PxABCB1*, *PxABCC1-3*, and *PxABCG1* genes[26–28]. In addition, two new *P. xylostella* strains designated APN1KO and APN3aKO were established by knocking out *PxAPN1* and *PxAPN3a* from DBM1Ac-S strain by CRISPR/Cas9 genome editing system, respectively. All *P. xylostella* strains were reared *en masse* on Jing Feng No. 1 cabbage (*Brassica oleracea* var. *capitata*) at 25 °C with 65% relative humidity (RH) and a 16:8 (light:dark) photoperiod, and adults were supplied with a 10% honey/water solution.

**Toxin preparation and bioassay**. Cry1Ac protoxin was prepared from Btk strain HD-73 and both purified Cry1Ac protoxin and trypsin-activated Cry1Ac toxin were quantified using Bradford's method with bovine serum albumin (BSA) as a standard[67]. Toxicity of Cry1Ac protoxin in 72 h bioassays with *P. xylostella* larvae was assessed using a leaf-dip method[65]. Briefly, 10 third-instar larvae were tested on each of seven toxin concentrations and bioassays were replicated in quadruplicate. The median lethal concentrations ($LC_{50}$) values and 95% confidence limits (CL) were calculated by probit analysis using POLO Plus 2.0 statistical software (LeOra Software). $LC_{50}$ values without overlapping among 95% CL were considered as significantly different.

**RNA extraction, cDNA synthesis, and gDNA isolation**. Different *P. xylostella* samples were collected and homogenized in TRIzol reagent (Invitrogen), and the total RNA was extracted, purified, and quantified according to the manufacturer's protocol. First-strand cDNA was then prepared either with the PrimeScript II 1st strand cDNA Synthesis Kit (TaKaRa) for gene cloning or with the PrimeScript RT kit (containing gDNA Eraser, Perfect Real Time) (TaKaRa) for qPCR detection following the manufacturer's recommendations. Genomic DNA (gDNA) samples of fourth-instar DBM1Ac-S larvae were isolated for gene cloning using a TIANamp Genomic DNA Kit (TIANGEN), and the individuals were fully homogenized with an electric pestle in 1.5 ml centrifuge tubes before gDNA extraction according to the manufacturer's instructions. Finally, the prepared cDNA and gDNA samples were used immediately or stored at −20 °C until used.

**Gene identification, cloning, and sequencing**. To identify all the PxAPN genes in *P. xylostella* genome, the amino acid sequences of the N-terminal conserved peptidase_M1 domain of M1 family aminopeptidases (Pfam ID: PF01433) in four known PxAPN proteins retrieved from GenBank database (PxAPN1, AAB70755; PxAPN2, CAA66467; PxAPN3b, AAF01259; PxAPN5, CAA10950) were used as

queries for a blastp search against the Diamondback moth Genome Database (DBM-DB, http://59.79.254.1/DBM/). Then, the putative CDS sequences of acquired genes were further in silico corrected with blastn search against our previous *P. xylostella* midgut transcriptome databases[32]. To obtain the full-length cDNA or partial gDNA sequences of all the M1 aminopeptidase genes, the PCR cloning strategies and the gene-specific primers designed by the Primer Premier 5.0 software (Premier Biosoft) are shown in Supplementary Table 6. The PCR reactions (25 µl) using cDNA or gDNA as template were conducted for 35 cycles with the following parameters: denaturing at 94 °C for 30 s, annealing at optimized 50–60 °C (depending on the primers) for 45 s, and extension at 72 °C for 1–4 min based on the PCR product size; and a final extension of 10 min at 72 °C using LA Taq polymerase with general or high GC buffers (TaKaRa). The obtained amplicons were purified, subcloned, and sequenced. After large-scale cloning and sequencing (20 clones from two independent cDNA batches), the full-length cDNA sequences of differential expressed PxAPN genes from DBM1Ac-S and NIL-R were compared to detect potential sequence variations.

**Bioinformatic analysis**. Gene sequence assembly, multiple sequence alignment, and exon–intron analysis were conducted using DNAMAN 9.0 (Lynnon BioSoft). The deduced protein sequences were obtained by Translate (https://web.expasy.org/translate/), and the calculated isoelectric point (pI) and molecular weight (Mw) were predicted with Compute pI/Mw (http://ca.expasy.org/tools/pi_tool.html). The N-terminal signal peptide was identified using SignalP 4.1 (http://www.cbs.dtu.dk/services/SignalP/). Two site prediction servers (big-PI Predictor: http://mendel.imp.ac.at/sat/gpi/gpi_server.html and GPI-SOM: http://gpi.unibe.ch/) were used to predict potential GPI-modification sites. Transmembrane helices were predicted using the TMHMM Server v. 2.0 (http://www.cbs.dtu.dk/services/TMHMM-2.0/). The presence of N- and O-glycosylation sites on the protein sequences were respectively predicted with the NetNGlyc 1.0 (http://www.cbs.dtu.dk/services/NetNGlyc/) and NetOGlyc 4.0 (http://www.cbs.dtu.dk/services/NetOGlyc/) servers. The conversed protein motifs were displayed by WebLogo 3 program (http://weblogo.threeplusone.com/create.cgi).

**Phylogenetic analysis**. Full-length protein sequences of all the current available lepidopteran PxAPN and other M1 aminopeptidase genes were retrieved and corrected from GenBank or other genome databases (Supplementary Table 2). To determine the phylogenetic relationship of these genes, sequence alignments were conducted with Clustal W in MEGA 7.0 (http://www.megasoftware.net/). The high-quality unrooted phylogenetic tree was then constructed using the maximum likelihood (ML) method based on the model optimized using the Bayes Information Criterion with "complete deletion" as the gaps/missing data treatment and 1000 bootstrap replications.

**qPCR analysis**. The gene transcript levels were quantified by real-time quantitative PCR (qPCR) detection processed in the QuantStudio 3 Real-Time PCR System (Applied Biosystems)[27]. Briefly, gene-specific primers of the PxAPN genes (Supplementary Table 6) with high amplification efficiencies (95–100%) were selected and used in qPCR reactions with 2.5× SYBR Green MasterMix Kit (TIANGEN) following the manufacturer's instructions. The qPCR program included an initial denaturation at 94 °C for 6 min, followed by 40 cycles of amplification at 94 °C for 30 s, 50–60 °C (depending on the primers) for 30 s, and 72 °C for 35 s. Relative expression levels were calculated using the $2^{-\Delta\Delta Ct}$ method and normalized to the ribosomal protein *L32* (*RPL32*) gene (GenBank accession no. AB180441).

**BBMV preparation**. Approximately 2000 fourth-instar larval midgut tissues from each *P. xylostella* strain were dissected in cold MET buffer [17 mM Tris–HCl (pH 7.5), 5 mM EGTA, 300 mM mannitol] plus protease inhibitors (1 mM PMSF). Midgut brush border membrane vesicles (BBMV) were prepared by the differential magnesium precipitation method and quantified using Bradford's method with BSA as a standard[26], and then flash-frozen and kept in aliquots at −80 °C until used. Between 5- and 8-fold enrichment in specific APN activity using L-leucine-*p*-nitroanilide (Leu-*p*-NA) (Sigma) as substrate was detected compared to initial midgut homogenates.

**Western blots**. Antibodies of PxAPN1 and PxAPN3a proteins used for Western blots were generated from synthetic peptides (Jiaxuan Biotech) derived from respective specific amino acid sequences [952]PPVDTTPDMTPPQP[965] and [977]DPPPEVDTETTPQP[990], and other specific antibodies targeting p38, JNK, ERK, and β-actin were commercially purchased (Supplementary Table 3). The protein and protein phosphorylation levels of target proteins were detected by Western blots using β-actin as an internal control. Midgut proteins were separated by sodium dodecyl sulfate-polyacrylamide gel electrophoresis and electrotransferred onto Immobilon-P membranes (Merck Millipore). The membranes were then blocked with blocking buffer containing BSA (CWBIO) at 25 °C for 1 h and incubated with the appropriate primary antibody (Supplementary Table 3) at 4 °C overnight, followed by incubation with goat anti-rabbit horseradish peroxidase-conjugated secondary antibody (1:5000, CWBIO) (Supplementary Table 3). The protein bands were visualized using the SuperSignal West Pico Chemiluminescent

Substrate (Thermo Fisher Scientific), and the images were captured by the Tanon-5200 Chemiluminescent Imaging System (Tanon). Densitometric analysis of the protein bands was performed using ImageJ v.1.51 software (http://rsbweb.nih.gov/ij/), and the relative band intensities were calculated based on densitometric ratios between target proteins and β-actin with the band intensities in the DBM1Ac-S BBMV samples considered as 1.

**Heterologous expression.** The recombinant PxAPN-GFP fusion proteins were in vitro transiently expressed in Sf9 cells[26]. The Sf9 cells were harvested, lysed and quantified, and specific APN activity assays and Western blots were performed. Enzymatic activity was detected using Leu-p-NA (Sigma) as substrate, with one enzymatic unit (U) being defined as the amount of the enzyme that would catalyze the production of the chromogenic product from 1 μmol specific substrate per min and per mg of total Sf9 cell proteins at 37 °C. Western blots were performed as mentioned above with mouse monoclonal anti-GFP antibody (1:1000, Abcam) using β-actin as an internal control. For immunolocalization of Cry1Ac toxin binding to Sf9 cells expressing PxAPN proteins, the transfected Sf9 cells were incubated with Cry1Ac toxin and fixed in ice-cold 4% paraformaldehyde. After blocking with 1% BSA, the cells were probed with rabbit polyclonal anti-Cry1Ac primary antibody (1:1000) and goat anti-rabbit secondary antibody conjugated with Alexa Fluor 555 (1:1000, Abcam). The treated cells were examined under a LSM 700 confocal laser scanning microscope (Carl Zeiss). For cytotoxicity assays, WST-8 in the Cell Counting Kit-8 (CCK-8) (Dojindo) was used to perform a CCK-8 assay following the manufacturer's recommendations. The absorbance was measured at 450 nm after 24 h incubation with Cry1Ac toxin, and the relative cell viability was calculated in relation to untreated Sf9 cells, which were defined as 100%. In addition, the morphological changes of Cry1Ac-treated Sf9 cells were also observed under a Zeiss LSM 700 microscope.

**RNA interference.** RNAi-induced gene silencing and subsequent functional assays were performed in vivo in *P. xylostella*[26–28,65]. Briefly, specific dsRNA was generated in vitro using the T7 Ribomax Express RNAi System (Promega) and subsequently mixed with an equal volume of Metafectene PRO transfection reagent (Biontex) before microinjection. Microinjection of dsRNA into newly molted third-instar *P. xylostella* larvae was conducted using the Nanoliter 2000 microinjection system (World Precision Instruments). Silencing effects were tested by qPCR and leaf-dip bioassays were performed to detect the larval toxic response to Cry1Ac protoxin post-RNAi.

**CRISPR/Cas9 experiment.** The CRISPR/Cas9 genome editing system has been recently successfully applied to efficiently probe gene functions in diverse insects[68], so CRISPR/Cas9-mediated gene knockout and subsequent functional validation were performed in vivo in *P. xylostella*[69]. Briefly, gene-specific sgRNA target sequences were designed in exon 12 of *PxAPN1* and exon 13 of *PxAPN3a* (Supplementary Table 6). In vitro transcription was conducted to generate sgRNAs with the MEGAscript Transcription Kit (Ambion) following the manufacturer's instruction. The mixtures of sgRNA (150 ng/μl) and Cas9 (300 ng/μl) were microinjected into fresh preblastoderm-stage *P. xylostella* eggs using FemtoJet 4i and InjectMan 4 Microinjection System (Eppendorf). Nondestructive genotyping of CRISPR/Cas9-induced indel mutations around the sgRNA target site was performed using PCR amplification with gDNA samples prepared from exuviates of individual fourth-instar larvae as templates, followed by direct sequencing and TA cloning. The germline transformation and mutation screening strategy was developed and optimized to construct stable homozygous mutant strains of both the *PxAPN1* and *PxAPN3a* genes designated APN1KO and APN3aKO, respectively (Supplementary Table 6). Leaf-dip bioassays were conducted to determine whether knockout of *PxAPN1* and *PxAPN3a* genes affects susceptibility to Bt Cry1Ac protoxin, and genetic complementation tests were further performed to characterize the Cry1Ac resistance phenotype in both knockout strains.

**Genetic linkage analysis.** The near-isogenic resistant NIL-R and susceptible DBM1Ac-S strains were used for genetic linkage analysis[26]. Briefly, a single-pair cross was conducted between a NIL-R male and a DBM1Ac-S female to generate the F1 progeny. Reciprocal crosses between single-pair F1 and NIL-R moths were performed to acquire two different backcross families. The resulting progenies from both backcross families were reared on control (cabbage) or experimental (cabbage with 20 mg/L of diagnostic Cry1Ac protoxin dose) diets, and single robust survivors were used for RNA extraction and cDNA synthesis as mentioned above. Linkage between gene expression levels or hormone titers and Cry1Ac resistance trait was tested using qPCR detection as described above.

**Hormone extraction and detection.** For JH and 20E extraction, *P. xylostella* samples were weighed and homogenized in glass homogenizers with 2 ml of ice-cold methanol/ether (1:1, v/v) or 75% methanol respective containing 0.1 ng JH analog methoprene (Dr. Ehrenstorfer GmbH; 98.5%) or 22S, 23S-homobrassinolide (Yuanye Biotech; 95%) as internal standards. The resulting homogenates were then vortexed vigorously and centrifuged three times for 10 min at 4500×g at 4 °C with 2 ml of *n*-hexane or 75% methanol to collect the hexane (upper) phase or the supernatant in brown siliconized glass tubes. Subsequently, the combined extracts

were dried completely under a pressure blowing concentrator and dissolved in 500 μl mobile phase (70% methanol for JH and 50% methanol for 20E). The final solution was filtered with 0.22 μm membrane before being subjected to chromatographic analysis.

Mixed calibration standards of 0.01, 0.1, 0.5, 1, 2, 5, and 10 ng/ml concentrations were prepared with 70% methanol for JHs [JH II (SciTech; 78%); JH III (Sigma; ≥65%)] and 50% methanol for 20E (Selleck Chemicals; 99.25%). Each solution contained 50 ng methoprene or 0.1 ng 22S, 23S-homobrassinolide as internal standards, respectively. Calibration curves were constructed by plotting the relative peak area ratios (the peak area of hormones/the peak area of their internal controls) versus their concentrations, and the regression lines were calculated using a weighted factor $(1/y)$ least-squares linear regression model. The final constructed calibration curves of JH II, JH III, and 20E were linear over concentration ranges with perfect regression correlation coefficients $(R^2 = 1)$, indicating the specificity of the constructed hormone detection methods (Supplementary Fig. 6).

To accurately quantify the JH and 20E titers, UPLC–MS/MS analysis was conducted using the ACQUITY UPLC I-Class/Xevo TQ-S micro System (Waters) equipped with an ACQUITY UPLC BEH C18 column (2.1 mm × 50 mm, 1.7 μm particle size). The chromatographic separation of JH and 20E was carried out using binary gradient program between methanol (mobile phase A) and water containing 0.1% formic acid (mobile phase B). For 20E detection, the gradient started with 70% of component B: water (0.1% formic acid) for 1 min and then decreased to 0% of component B within 5 min. This gradient was kept for 8 min, and then increased to 70% of component B within 0.5 min. The total run time was 10 min, and an equilibration step of 1.5 min was included. For JH detection, the gradient started with 30% of component B: water (with 0.1% formic acid) for 2 min and then decreased to 15% of component B within 2 min. This gradient was kept for 2 min, and then increased to 30% of component B within 2 min. The total run time was 9 min, and an equilibration step of 1 min was included. The flow rate of the mobile phase was constant at 0.2 ml/min (injection volume: 10 μl) and the column temperature was 40 °C. To enhance the selectivity and sensitivity of the MS detection, the mass spectrometer was processed under electrospray ionization positive ion mode with MRM of two transitions per compound (precursor ions/product ions). The source parameters under MRM detection were optimized for JH and 20E: source temperature 150 °C, desolvation temperature 400 °C, cone gas flow 50 L/h, desolvation gas flow 800 L/h, source voltage 1760 V, core voltage 62 V. To enhance the selectivity and sensitivity of the MS detection, the MRM transition parameters applied for JH and 20E and their respective internal controls methoprene and 22S, 23S-homobrassinolide with regard to the transitions from precursor to product ions were also optimized and shown in Supplementary Table 5. The system operation, data acquisition, and analysis were performed using the MassLynx V4.1 software (Waters).

During the UPLC–MS/MS detection of JH and 20E titers, the limits of detection (LOD), defined as the lowest concentration that the analytical process can differentiate from background levels, were estimated for a signal-to-noise ratio (SNR, also called the *S/N* ratio) of 3 from the chromatograms of the detected samples, while limits of quantification (LOQ), defined as the lowest concentration that the analytical process can quantify from background levels, were estimated for a SNR of 10 from the chromatograms of the detected samples. Of particular note, the LOD and LOQ are very important threshold values to consider when measuring 20E titers since the 20E levels in the larval molting stages are very low (Supplementary Fig. 7).

**Hormone assays.** Hormone treatments were performed using newly molted third-instar *P. xylostella* larvae. 20E and methoprene were dissolved in ethanol (20 mg/ml) and diluted with ice-cold insect Ringer's solution (130 mM NaCl, 0.5 mM KCl, 0.1 mM CaCl₂) before microinjection. The MAPK inhibitors SB203580 (specific p38 inhibitor, Merck Millipore), SP600125 (specific JNK inhibitor, Merck Millipore) and PD0325901 (specific MEK/ERK inhibitor, TargetMol) were dissolved in dimethyl sulfoxide (DMSO, Sigma Aldrich). Specific dsRNA targeting *PxMAP4K4* (dsPxMAP4K4) was prepared as described in the RNAi section. The detection time of hormone-induced effects on gene expression and the quantities of hormones injected was optimized in preliminary microinjection experiments (Supplementary Fig. 8). Subsequently, newly molted third-instar DBM1Ac-S larvae were starved for 6 h and anesthetized for 30 min on ice, 70 nl solutions containing 20E (2 ng), methoprene (200 ng), mixture of 20E (2 ng), and methoprene (200 ng) or the hormone mixture with 100 μM MAPK inhibitor cocktail and dsRNA combined with hormone treatments were respectively microinjected into some of these larvae using the Nanoliter 2000 microinjection system, whereas the remaining control larvae were injected with the equal volumes of buffer (diluted ethanol and DMSO), with the final concentration of ethanol not exceeding 0.05%. More than 30 larvae were microinjected for each treatment, and these injected larvae were allowed to recover for about 3 h at room temperature and were then returned to normal rearing conditions, the final survivors from each control and treatment groups were used for the subsequent midgut dissection, qPCR analysis, and Western blots as described above. Meanwhile, leaf-dip bioassays were performed for 72 h using larvae at 48 h after hormone, MAPK inhibitors or dsRNA injection and Cry1Ac protoxin concentrations representing approximately the LC₅₀ (1 mg/L) or LC₂₀ (0.3 mg/L) value for non-injected DBM1Ac-S larvae. Bioassays were performed with 20 larvae in each group, and each bioassay replicated three times. Mortality in

control treatments was below 5% and bioassay data processing was as described above.

**Toxin induction assays**. Toxin induction of insect hormones was conducted using newly molted third-instar DBM1Ac-S and NIL-R larvae[26]. The DBM1Ac-S or NIL-R larvae were selected with 1 or 3500 mg/L Cry1Ac protoxin (respective LC$_{50}$ values for each strain) for 72 h as for the leaf-dip bioassay mentioned above, unselected larvae from both strains were used as control groups. After 72 h, sample preparation from both robust and weak survivors to detect hormone titers was undertaken as described above. Three independent assays were performed, and one-way ANOVA with Holm–Sidak's test (overall significance level = 0.05) was used to determine the significant statistical difference between control and treatment groups.

**Fitness cost analysis**. The effects of exogenous hormone and Cry1Ac protoxin treatments on subsequent developmental stages of surviving larvae to determine potential fitness costs were analyzed by comparing biological parameters including pupation percentage, pupal weight, pupation duration, eclosion percentage, adult longevity, oviposition duration, female fecundity, and egg hatchability. Larvae injected with buffer and unselected with toxin were used as negative controls in the hormone treatment and toxin induction experiments, respectively. All the larvae used in the test were fed on fresh cabbage leaves. Each treatment was replicated three times with 10 larvae per replicate. One-way ANOVA with Holm–Sidak's test (overall significance level = 0.05) was used to assess statistical significance of differences in these biological parameters related to fitness costs between control and treated groups.

**Reporting summary**. Further information on research design is available in the Nature Research Reporting Summary linked to this article.

## Data availability

The full-length cDNA sequences of all the cloned genes in this study have been deposited in the GenBank database (Accession Nos. MG873047–MG873063 and MH213067–MH213068). The authors declare that the data supporting the findings of this study are available within the paper and its Supplementary Information. The source data underlying Figs. 1c, d, 2b–f, 3, 4, 5b–f, 6, 7b–i, k–r, and 8 and Supplementary Figs. 1a, 2, 3a, b, 4, 6, and 8 are provided as a Source Data file.

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

## Acknowledgements

We thank Prof. Kaiyu Liu from the Central China Normal University for providing the pie2-GFP-N1 expression vector, and we also thank Prof. Xia Cui and Dr. Haijing Wang from our institute for the assistance with the UPLC–MS/MS experiment. This work was supported by the National Natural Science Foundation of China (31630059 and 31701813), the Central Public-interest Scientific Institution Basal Research Fund (IVF-BRF2020015), the Beijing Key Laboratory for Pest Control and Sustainable Cultivation of Vegetables, and the Science and Technology Innovation Program of the Chinese Academy of Agricultural Sciences (CAAS-ASTIP-IVFCAAS).

## Author contributions

Z.G., X.Z., N.C., and Y.Z. designed the research. Z.G., S.K., D.S., L. Gong, J.Z., J.Q., L. Guo, L.Z., Y.B., F.Y., Q.W., and S.W. performed the experiments. Z.G. analyzed the data. Z.G., N.C., X.Z., and Y.Z. wrote and revised the manuscript.

## Competing interests

The authors declare no competing interests.
