## [Peer Review File · Nature Communications]

Reviewers' Comments:

Reviewer #1:

Remarks to the Author:

This paper shows that two isoforms of the 12 aminopeptidases N in the diamondback moth larvae (APN1 and APN3a) are the primary receptors for the *Bacillus thuringiensis* toxin and that their downregulation by RNAi or their loss by CRISPR-induced mutation leads to resistance to the Bt toxin. They also showed in the resistant strain (NIL-R) that the transcript levels of these two genes were reduced. Moreover, suppression of MAP4K4 by RNAi led to an increase in these two enzyme mRNAs as well as several other enzymes involved in susceptibility to the toxin. This part of the study appears to be competently done and well controlled. By contrast, the part of the study concerned with the effects of the two developmental hormones, 20-hydroxyecdysone (20E) and juvenile hormone (JH), has many problems. Until these are corrected, it is not possible to evaluate critically their hypothesis on the effect of hormones on the response to Bt toxin. These problems are the following:

(1) It is not sufficient to take one sample per larval stage and assay it for both 20E and JH. In lepidopteran larvae the JH titers fluctuate during the intermolt feeding stages (prior to the final feeding stage), then usually decrease during the molt. In contrast, the 20E titer is low during the intermolt phase, then increases to a peak driving the molt, falling before ecdysis to the next stage. Hence, one sample per stage does not reflect accurately these titers. Also, it is not appropriate to connect the preadult titers with the individual male and female adult titers.

(2) It is not clear what a "typical" first to fourth instar larva is. Even for the prepupal stage one needs to know whether it is early when the ecdysteroid titer is increasing or later in the pharate pupa stage when ecdysteroid titer is decreasing. Although the developmental ages of the green and yellow pupae are unclear, the fairly high level of JH in these stages shown in Figure 5b is very unexpected for a lepidopteran and should be commented upon.

(3) For the comparisons in Figure 5 c,d,g,h, one must use 4th instar larvae of comparable ages, rather than just "typical" larvae.

(4) It is important in the hormone injection experiments to indicate the developmental stage of the larva, both when injected and when assessed 48 hr later (Fig. 5e). It seems that "early" 3rd instar larvae were injected (line 248), but mRNA levels in "typical" 4th instar larvae were assessed (line 1019). Does "early" mean the day of ecdysis to the 3rd instar? Was there any difference in times of ecdysis to the 4th instar among the methoprene-injected, the 20E-injected, and those injected with both hormones?

(5) Although the text on line 266 says that $P < 0.05$ for the difference between the 20E treatment and the 20E+methoprene treatment, in Figure 5f, both have the same letter c indicating that there is no significant difference. Please reconcile this

difference.

(6) When were the animals in Figure 6a-e treated with the hormones? In Figure 6e, are the ones treated with either or both hormones as viable as adults as the controls? Do the ones treated with methoprene with or without 20E show any developmental anomalies as adults?

Minor corrections necessary:

- 1) Abstract, lines 29 and 31; and lines 227,357, 377: "Endocrine hormones" is not a proper designation. "Hormones" is sufficient. Hormones are secreted by endocrine glands or by neurosecretory cells into the blood to affect distant target organs. The two hormones studies here, ecdysone and JH, are secreted by endocrine glands.
- 2) line 122: ...xylostella41. Likewise...
- 3) line 153: Should be Figure 2a, not Supplementary Figure 2a.
- 4) line 317: ...sexta20,29. In....
- 5) lines 1035 and 1039: Pupal....
- 6) line 1036: ...a representative pupa from a treated larva (...)
- 7) line 1058: ...increased...

Reviewer #2:

Remarks to the Author:

This manuscript describes the characterization of the aminopeptidase (APN) gene family and its relation to the evolution of resistance to Cry1Ac toxin in *Plutella xylostella*. Four different *P. xylostella* resistant populations showed a decreased expression of two APN isoforms PxAPN1 and PxAPN3a while an increased expression of PxAPN5 and PxAPN6 was observed. Expression of both PxAPN1 and PxAPN3a in Sf9 cells and mutation by CRISP/CAS of both genes confirmed that both APN isoforms are functional receptors of Cry1Ac. Silencing of MAP4K4 in a resistant strain (NiIR), which was previously shown to have increased MAP4K4 expression that correlated with decreased ABCC receptors expression, the authors showed that PxMAP4K4 is involved in regulation of APN differential expression in the NiIR strain. Furthermore a link between two developmental hormones as JH and 20E and PxMAP4k4 was established by showing that all R strains have increased titers of both hormones while exogenous application of these hormones confirmed that JH and 20E were involved in PxMAP4K4 expression. This is an interesting manuscript that establish for the first time a link between hormone triggering MAPK signal pathway that results in lowering Cry1Ac receptors expression and resistance to Cry1Ac. However it is difficult to establish if the increased titers of both hormones in the resistant strains is a cause or effect of resistance since 4 different Px resistant strains that likely have different resistant mechanisms have a similar phenotype regarding hormone titers. This has to be discussed and more information on the

resistant mechanism of the different resistant strains needs to be provided.

Additional points

1. Line 122-125. Define the differences between these Cry1Ac resistant strains. Are they affect in different genes? What are the resistant Ratios of the different strains? etc.
2. Figure 1D. What anti-APN antibody were used for western blots? Are these antibodies specific and do not cross react with other APN isoforms. This has to be clear in the results section.
3. Silencing of either APN5 or APN6 as controls to determine the specificity of APN1 and APN3 in Cry1Ac toxicity.
4. Hormone titers are increased in R strains but is this the cause or effect? Again are all the Cry1Ac R strains similar genetically? What is the effect of silencing MAPK4 when S strain is treated with hormones? Restores susceptibility?
- 5, MAPK signaling pathways are regulated by phosphorylation cascade rather than by transcriptional regulation. Short time exposure to both JH or 20E should have a consequence in phosphorylation of different components of the signaling pathway. Antibodies that recognize phosphorylated isoforms of MAPK38 or JNK are available commercially. It could be interesting to test this or discuss the possible regulation of phosphorylation activation by both hormones.
6. How are the titers of both hormones increased in all R strains which are not genetically similar? Are the high hormone titers genetically linked to Cry1Ac resistant in the R strains? At least the NilR strain could be characterized as the APN expression genetic linkage.

Response to Reviewers

Reviewer #1

This paper shows that two isoforms of the 12 aminopeptidases N in the diamondback moth larvae (APN1 and APN3a) are the primary receptors for the *Bacillus thuringiensis* toxin and that their downregulation by RNAi or their loss by CRISPR-induced mutation leads to resistance to the Bt toxin. They also showed in the resistant strain (NIL-R) that the transcript levels of these two genes were reduced. Moreover, suppression of MAP4K4 by RNAi led to an increase in these two enzyme mRNAs as well as several other enzymes involved in susceptibility to the toxin. This part of the study appears to be competently done and well controlled. By contrast, the part of the study concerned with the effects of the two developmental hormones, 20-hydroxyecdysone (20E) and juvenile hormone (JH), has many problems. Until these are corrected, it is not possible to evaluate critically their hypothesis on the effect of hormones on the response to Bt toxin. These problems are the following:

We thank the reviewer for their positive comments concerning the aminopeptidase N study. Although the hormone section will be of more general interest, the first section is nonetheless of importance in understanding the 1) increasing problem of resistance to Bt products and 2) involvement of insect hormones in Bt resistance.

1. It is not sufficient to take one sample per larval stage and assay it for both 20E and JH. In lepidopteran larvae the JH titers fluctuate during the intermolt feeding stages (prior to the final feeding stage), then usually decrease during the molt. In contrast, the 20E titer is low during the intermolt phase, then increases to a peak driving the molt, falling before ecdysis to the next stage. Hence, one sample per stage does not reflect accurately these titers. Also, it is not appropriate to connect the preadult titers with the individual male and female adult titers.

We have collected samples from both intermolt feeding and molting stages and redetected the JH and 20E titers each day during all the developmental stages of *P. xylostella* to precisely reflect the fluctuation of both insect hormone titers referred to by the reviewer. We have also unified the naming of developmental stages to better describe the hormone titer changes. We have reanalyzed the new hormone data, redrawn the figure (Fig. 5b) and revised the corresponding sections to address the reviewer's concerns.

2. It is not clear what a "typical" first to fourth instar larva is. Even for the prepupal stage one needs to know whether it is early when the ecdysteroid titer is increasing or later in the pharate pupa stage when ecdysteroid titer is decreasing. Although the developmental ages of the green and yellow pupae are unclear, the fairly high level of JH in these stages shown in Figure 5b is very unexpected for a lepidopteran and should be commented upon.

We took samples from newly molted, one-day-old feeding larvae. This has been clarified in the text, as mentioned above, we have also resampled larvae at various defined stages to not only show the pattern of hormone fluctuation described by the reviewer but also to reference hormone

titers at the sampling stage. Fig. 5b has been redrawn with these new data.

3. For the comparisons in Figure 5 c,d,g,h, one must use 4th instar larvae of comparable ages, rather than just “typical” larvae.

As mentioned above, we have clarified the stage at which larvae were sampled – one-day-old feeding 4th instar. To further reassure the reader, we have taken multiple samples throughout the 4th instar and found that hormone titers were relatively constant during the feeding stages. Thus, we have confidence that comparisons between one-day-old feeding larvae from different strains/treatments are valid.

4. It is important in the hormone injection experiments to indicate the developmental stage of the larva, both when injected and when assessed 48 hr later (Fig. 5e). It seems that “early” 3rd instar larvae were injected (line 248), but mRNA levels in “typical” 4th instar larvae were assessed (line 1019). Does “early” mean the day of ecdysis to the 3rd instar? Was there any difference in times of ecdysis to the 4th instar among the methoprene-injected, the 20E-injected, and those injected with both hormones?

Following the reviewer’s comments, we have clarified that “early” meant “newly molted”, and as mentioned above, clarified that “typical” meant one-day-old feeding. As the reviewer correctly anticipated, there were indeed some differences in times of ecdysis to the 4th instar among the methoprene-injected, the 20E-injected, and those injected with both hormones. Thus, we can’t state that larvae were at the same stage 48 h later. However, for each experiment, we only sampled feeding stages and so having shown that there were no significant differences in hormone titers between 3rd and 4th instar feeding stages (Fig. 5c, d), we are confident that the comparisons made are valid.

5. Although the text on line 266 says that $P < 0.05$ for the difference between the 20E treatment and the 20E+methoprene treatment, in Figure 5f, both have the same letter c indicating that there is no significant difference. Please reconcile this difference.

We apologize for this error, and changed the letter “c” into “d” in the figure to indicate the significant difference.

6. When were the animals in Figure 6a-e treated with the hormones? In Figure 6e, are the ones treated with either or both hormones as viable as adults as the controls? Do the ones treated with methoprene with or without 20E show any developmental anomalies as adults?

We have clarified that both the hormone and toxin induction treatments were performed using newly molted third-instar larvae. We have added further biological parameters to Fig. 7 including adult longevity, oviposition duration, female fecundity and egg hatchability in both assays to answer the question about adult developmental anomalies. As expected, single hormone treatment and weak individuals in the toxin treatments had obvious fitness costs throughout their life cycle. In contrast, dual hormone treatment and robust individuals in the toxin treatments didn’t display

obvious developmental anomalies compared to the controls.

Minor corrections necessary:

1. Abstract, lines 29 and 31; and lines 227,357, 377: “Endocrine hormones” is not a proper designation. “Hormones” is sufficient. Hormones are secreted by endocrine glands or by neurosecretory cells into the blood to affect distant target organs. The two hormones studies here, ecdysone and JH, are secreted by endocrine glands.

Corrected.

2. line 122: ...xylostella41. Likewise...

Corrected.

3. line 153: Should be Figure 2a, not Supplementary Figure 2a.

Corrected.

4. line 317: ...sexta20,29. In....

Corrected.

5. lines 1035 and 1039: Pupal....

Corrected.

6. line 1036: ...a representative pupa from a treated larva (...)

Corrected.

7. line 1058: ...increased...

Corrected.

Reviewer #2

This manuscript describes the characterization of the aminopeptidase (APN) gene family and its relation to the evolution of resistance to Cry1Ac toxin in *Plutella xylostella*. Four different *P. xylostella* resistant populations showed a decreased expression of two APN isoforms P_xAPN1 and P_xAPN3a while an increased expression of P_xAPN5 and P_xAPN6 was observed. Expression of both P_xAPN1 and P_xAPN3a in Sf9 cells and mutation by CRISP/CAS of both genes confirmed that both APN isoforms are functional receptors of Cry1Ac. Silencing of MAP4K4 in a resistant strain (NiLR), which was previously shown to have increased MAP4K4 expression that correlated with decreased ABCC receptors expression, the authors showed that P_xMAP4K4 is involved in regulation of APN differential expression in the NiLR strain. Furthermore a link between two developmental hormones as JH and 20E and P_xMAP4k4 was established by showing that all R strains have increased titers of both hormones while exogenous application of these hormones confirmed that JH and 20E were involved in P_xMAP4K4 expression. This is an interesting manuscript that establish for the first time a link between hormone triggering MAPK signal pathway that results in lowering Cry1Ac receptors expression and resistance to Cry1Ac. However it is difficult to establish if the increased titers of both hormones in the resistant strains is a cause or effect of resistance since 4 different P_x resistant strains that likely have different resistant mechanisms have a similar phenotype regarding hormone titers. This has to be discussed and more information on the resistant mechanism of the different resistant strains needs to be provided.

We thank the reviewer for their precise summary of our work. As to the reviewer's concern about the difficulty of establishing whether the observed increased hormone titers in the four resistant strains is a cause or effect of resistance, we have attempted to clarify this below and in the manuscript. The observed increase in hormone titers in all the resistant strains led us to test the possibility of a causal effect in the near-isogenic resistant (NIL-R) and susceptible (DBM1Ac-S) strains. We believe that the various experiments that we undertook on these two strains demonstrate a causal link. Although we cannot translate this causal relationship directly to other strains, given that our previous studies in PLoS Genetics etc. had found the same resistance mechanism among the four resistant strains (involving the MAPK signaling pathway *trans*-regulated differential alteration of multiple midgut receptors and non-receptor paralogs), we believe that there is a strong likelihood that the findings for NIL-R also apply to the other strains. We have discussed this fact and have also provided more information on the resistance mechanisms of the different strains as per the reviewer's suggestions.

Additional points

1. Line 122-125. Define the differences between these Cry1Ac resistant strains. Are they affected in different genes? What are the resistant Ratios of the different strains? etc.

As described above, although the four tested resistant strains are geographically different strains, we have found the same MAPK-mediated resistance mechanism, and the same midgut receptor and non-receptor paralogous genes being involved in this novel resistance mechanism in these resistant strains. We have provided information detailing the resistance ratios and the involved midgut resistance genes in these strains in the Introduction as well as Methods sections in the

previous version of our manuscript, and we have also further revised these sections to make it more clearly.

2. Figure 1D. What anti-APN antibody were used for western blots? Are these antibodies specific and do not cross react with other APN isoforms. This has to be clear in the results section.

We have now provided further information of the specific anti-APN antibodies used in the Western Blots section of the Methods section. We have also added a new Supplementary Table 3 to list the details of all the antibodies used in this study.

3. Silencing of either APN5 or APN6 as controls to determine the specificity of APN1 and APN3 in Cry1Ac toxicity.

Thank you for suggesting this important experiment. We have conducted this RNAi experiment and found that, unlike with APN1 and APN3a, silencing of APN5/6 has no effect on susceptibility to Cry1Ac thus supporting both our heterologous expression data and indeed the hypothesis that overexpression of APN5/6 may compensate for loss of APN1/3a. These findings have been added to the text and the data presented in Supplementary Fig. 4.

4. Hormone titers are increased in R strains but is this the cause or effect? Again are all the Cry1Ac R strains similar genetically? What is the effect of silencing MAPK4 when S strain is treated with hormones? Restores susceptibility?

As mentioned above, we are only claiming to show a causal link between hormone titer and resistance in the NIL-R strain. This one provided a good model since it is near isogenic to the susceptible control strain. The proposed experiment to investigate the interplay between hormone exposure and *MAP4K4* silencing was undertaken and the results shown in Fig. 6d, e. The data do indeed show that the increase in susceptibility observed when the strain is treated with dsMAP4K4 is reversed when it is treated with both dsRNA and the hormone combination.

5. MAPK signaling pathways are regulated by phosphorylation cascade rather than by transcriptional regulation. Short time exposure to both JH or 20E should have a consequence in phosphorylation of different components of the signaling pathway. Antibodies that recognize phosphorylated isoforms of MAPK38 or JNK are available commercially. It could be interesting to test this or discuss the possible regulation of phosphorylation activation by both hormones.

We thank the reviewer for this suggestion to further test our hypothesis. Using commercial antibodies against total and phosphorylated forms of p38, ERK and JNK, we demonstrated that exposure to 20E or the hormone combination does result in an increase in the phosphorylated forms of all three of these downstream components. In contrast, exposure to methoprene resulted in a small reduction in phosphorylation. In addition, we also performed an important experiment to investigate the effect of specific inhibitors of these downstream kinases on toxin susceptibility and found that the use of inhibitors almost entirely blocked the hormone induced loss of

susceptibility. This further strengthened our hypothesis that the hormones were acting via the MAPK signaling cascade. These new data have been added to Fig. 6.

6. How are the titers of both hormones increased in all R strains which are not genetically similar? Are the high hormone titers genetically linked to Cry1Ac resistant in the R strains? At least the NilR strain could be characterized as the APN expression genetic linkage.

The reviewer poses an interesting question, but as mentioned above, considering that all the resistant strains have the same resistance mechanism, we can reasonably speculate that similar genetic changes are responsible for the resistance phenotype are present in all strains. The second good suggestion was tested for NIL-R and genetic linkage was found between increased titers of 20E and resistance, in contrast, there was no linkage between titers of JHII and resistance. Thanks to this excellent suggestion, we have been able to further refine our model to one in which 20E determines resistance and JHII modules the effect of the former hormone by minimizing the fitness costs associated with 20E-induced repression of midgut protein expression. These new data have been incorporated into Fig. 8.

Reviewers' Comments:

Reviewer #1:

Remarks to the Author:

The authors have significantly revised the paper and shown that the near isogenic NIL-R resistant strain has reduced levels of the two specific aminopeptidases N that act as receptors for the Bt toxin which thereby increases their resistance to the toxin. They then show that three enzymes in the MAPK pathway (JNK, Erk and p38) up-regulate other aminopeptidases N as well as other midgut genes to counter the virulence of the toxin. In turn, the MAPK cascade is regulated by the developmental hormones, juvenile hormone (JH) and ecdysteroids, primarily the latter.

In answer to my criticisms, they now have a good developmental series of both JH and ecdysteroid titers as well as interpretable hormone manipulation experiments. However, the data on the ecdysteroid titers show an unexpected peak of ecdysteroid levels on day 0 of each larval instar preceded by a very low titer in the feeding intermolt stage. One expects the ecdysteroid titer to peak some time before ecdysis to the next stage since it is ecdysone and 20-hydroxyecdysone (20E) that initiate and orchestrate the molt. In fact, in both *Manduca* and *Bombyx* where extensive and detailed ecdysteroid titers have been done, 20E must fall to low levels in order for the hormones governing ecdysis to be released (see Curtis et al., 1984, *J. Insect Physiol.* for experiments showing that ecdysteroid levels must be low for larval ecdysis to occur). Only the ecdysteroid titer during pupal-adult development matches that of other Lepidoptera and insects in general. It seems possible that the method used for extraction and liquid chromatography-mass spectrophotometric analysis of 20E also extracted some of the conjugated and inactive ecdysteroids. If these were not subsequently removed through Sep-pack or other prepurification, they may be converted to 20E during the analysis, thereby producing an abnormally high level of 20E. In *Drosophila*, CYP18A1 is the main ecdysteroid inactivation enzyme causing conversion of 20E to 20,26E and thence to 20Eoic acid, both of which are inactive (Guittard et al., *Dev. Biol.*, 2011). The expression of this enzyme peaks at the time of ecdysis to the 3rd instar and is induced by the 20E peak that triggers the molt. The authors should be aware of the types of the compounds that are present in whole body methanolic extracts and the influence that they can have on the levels of 20E analyzed (see the papers by Scalia et al., *Insect Biochem.* 17, 227- (1987) which discusses the problems when analyzing ecdysteroid levels in grasshopper eggs and the review by Lafont, R., et al. (2005). *Ecdysteroid chemistry and biochemistry*. In: Gilbert, L.I., Iatrou, K., Gill, S.S. (Eds.), *Comprehensive Molecular Insect Science*, vol. 3. Elsevier, Oxford, pp. 125–195). The ecdysteroid titers likely will all have to be done over using procedures for whole body extracts that allow one to only measure 20E itself.

Minor concerns that need attention:

1) In Fig. 5b, they show titers for days 0 and 1 of each larval instar and then connect all the points as if there was only 1 day in each of the first 3 instars and 2 days in the 4th instar. Yet in Fig. 5 C they have points for 0-36 h in L3 and for 0-72 hr in L4. Therefore, the lines should not be connected between the 3rd, 4th and pupa in Fig. 5b.

2) In the legend to figure 7, line 1134, one should indicate when the hormone was given as follows: "...treatment on day 0 of the 3rd larval instar." In the text describing this figure, they should indicate whether those larvae given 20E molted early to the 4th instar and to the pupa. One would expect that the small size of the resultant pupa is due to a premature molt to the 4th instar but that 4th instar larva is above the threshold size for metamorphosis so goes through a normal length 4th instar. Certainly one injection of 20E would not last into the 4th instar. More discussion is needed for the interpretation of these results.

Reviewer #2:

Remarks to the Author:

The authors performed additional experiments that in my point of view address all reviewers' comments. In particular they performed experiments to define if the higher titers of hormones are cause or effect of the resistance mechanism. The authors show that the increased susceptibility of larvae treated with dsMAP4K4 RNA is reversed by treatment with hormones and that high 20E-hormone titer is genetically linked to Cry1Ac resistance. Although the genetic basis for the high 20E-hormone titer still needs to be define, I believe that this study is a breakthrough in the role of insect hormones in triggering resistance to pathogen infection in insects.

Minor comments

1. The genetic basis of increase 20E-hormone titer in NIL-R strain till needs to be define. Are there any other examples of insect mutations that cause enhanced hormone titers in insects? What are the phenotypes? Discuss briefly in Discussion section.

2. PxAPN3a is slightly down regulated in all strains. If similar genetic basis for resistance is shared in all resistant strain how differences in APN expression of other APN's (e.g. PxAPN13, PxAPN2 etc.) are explained. Additional mutations? Brief discussion of the possible cause could be included in Discussion section.

Response to Referees Letter

Reviewers' comments:

Reviewer #1 (Remarks to the Author):

The authors have significantly revised the paper and shown that the near isogenic NIL-R resistant strain has reduced levels of the two specific aminopeptidases N that act as receptors for the Bt toxin which thereby increases their resistance to the toxin. They then show that three enzymes in the MAPK pathway (JNK, Erk and p38) up-regulate other aminopeptidases N as well as other midgut genes to counter the virulence of the toxin. In turn, the MAPK cascade is regulated by the developmental hormones, juvenile hormone (JH) and ecdysteroids, primarily the latter.

In answer to my criticisms, they now have a good developmental series of both JH and ecdysteroid titers as well as interpretable hormone manipulation experiments. However, the data on the ecdysteroid titers show an unexpected peak of ecdysteroid levels on day 0 of each larval instar preceded by a very low titer in the feeding intermolt stage. One expects the ecdysteroid titer to peak some time before ecdysis to the next stage since it is ecdysone and 20-hydroxyecdysone (20E) that initiate and orchestrate the molt. In fact, in both *Manduca* and *Bombyx* where extensive and detailed ecdysteroid titers have been done, 20E must fall to low levels in order for the hormones governing ecdysis to be released (see Curtis et al., 1984, *J. Insect Physiol.* for experiments showing that ecdysteroid levels must be low for larval ecdysis to occur). Only the ecdysteroid titer during pupal-adult development matches that of other Lepidoptera and insects in general. It seems possible that the method used for extraction and liquid chromatography-mass spectrophotometric analysis of 20E also extracted some of the conjugated and inactive ecdysteroids. If these were not subsequently removed through Sep-pack or other prepurification, they may be converted to 20E during the analysis, thereby producing an abnormally high level of 20E. In *Drosophila*, CYP18A1 is the main ecdysteroid inactivation enzyme causing conversion of 20E to 20,26E and thence to 20Eoic acid, both of which are inactive (Guittard et al., *Dev. Biol.*, 2011). The expression of this enzyme peaks at the time of ecdysis to the 3rd instar and is induced by the 20E peak that triggers the molt. The authors should be aware of the types of the compounds that are present in whole body methanolic extracts and the influence that they can have on the levels of 20E analyzed (see the papers by Scalia et al., *Insect Biochem.* 17, 227- (1987) which discusses the problems when analyzing ecdysteroid levels in grasshopper eggs and the review by Lafont, R., et al. (2005). *Ecdysteroid chemistry and biochemistry*. In: Gilbert, L.I., Iatrou, K., Gill, S.S. (Eds.), *Comprehensive Molecular Insect Science*, vol. 3. Elsevier, Oxford, pp. 125–195). The ecdysteroid titers likely will all have to be done over using procedures for whole body extracts that allow one to only measure 20E itself.

Although we have addressed the vast majority of reviewer's concerns in the revised manuscript, there is still one outstanding in the detection method of 20E titer during larval ecdysis. Specifically, the reviewer noticed that the 20E titers showed an unexpected peak on day 0 of each larval instar, which should be low for larval ecdysis to occur according to previous studies in both tobacco hornworm (*Manduca sexta*) and silkworm (*Bombyx mori*). The reviewer suspected that our method might not be able to distinguish between 20E and other conjugated and inactive ecdysteroids to paint an accurate picture of 20E levels during ecdysis.

Our hormone extraction and detection methods are consistent with previous insect hormone studies (see the Methods section in the manuscript where we have provided detailed hormone

extraction and detection methods to allow others to replicate our experiments). In particular, the advanced detection instrument (a Waters ACQUITY UPLC I-Class/Xevo TQ-S micro System equipped with an ACQUITY UPLC BEH C18 column) with the companion detection method (UPLC-MS/MS with multiple reaction monitoring (MRM) and internal controls can precisely detect JH and 20E titers in *P. xylostella* samples. The non-target conjugated and inactive ecdysteroids in the whole body extracts can be filtered out during the MRM process in positive ion mode with specific transition parameters of 20E and its corresponding internal control 22S, 23S-homobrassinolide during the UPLC-MS/MS analysis. From a technical standpoint, the MRM process includes several channels, it is composed of precursor ion and product ion (e.g., see the distinguishable ion data of 20E and its metabolites in Table 1 of Destrez et al., 2008, Agilent Technologies Technical Note, Table 1 of Hikiba et al., J. Chromatogr. B, 2013 and Table 2 of Venne et al., 2016, J. Chromatogr. A). When the function channels are set for 20E and its corresponding internal control 22S, 23S-homobrassinolide, the other compounds including the conjugated and inactive ecdysteroids will be eliminated. If we wanted, we could simultaneously detect and quantify these conjugated and inactive ecdysteroids in the samples if we also set their specific MRM transition parameters in the function channels during MS detection (see the distinguishable ion chromatograms of 20E and its metabolites in Fig. 1 of Hikiba et al., J. Chromatogr. B, 2013 and Figs. 2&4 of Venne et al., 2016, J. Chromatogr. A). Hence, those compounds don't influence the levels of 20E analyzed in this study. Moreover, in the reference the reviewer provided (Scalia et al. Insect Biochem., 1987, 17, 227–236), we can see that the authors actually used two different chromatography methods (different columns, reagents and procedures) to separately detect and analyze the conjugated and free ecdysteroids using HPLC under U.V. detection (not the more advanced and precise UPLC with tandem MS detection). That approach also lacked the structurally simulated internal control of ecdysteroids such as 22S, 23S-homobrassinolide used in our study and as suggested in Destrez et al., 2008, Agilent Technologies Technical Note). Consequently, we are confident that our method can accurately and specifically measure 20E titers.

However, in the process of assessing our procedures, we did uncover an issue that could have affected our results. For the sake of explanation, we need to mention three technical terms including the Signal-to-Noise Ratio (SNR, also called the S/N ratio), limit of detection (LOD) and limit of quantification (LOQ) during UPLC-MS/MS detection. Signal-to-Noise Ratio (SNR) has been a primary standard for evaluating the MS performance of chromatography systems including GC/MS and LC/MS. Noise is measured by the Root-Mean-Squared (RMS) value of the baseline over the selected time window, and the SNR is defined as the average over time of the peak signal divided by the RMS noise of the peak signal over the same selected time window (Wells et al., 2011, Agilent Technologies Technical Note). SNR is a common parameter used to specify the limit of detection (LOD) and limit of quantification (LOQ). Limits of detection (LOD), defined as the lowest concentration that the analytical process can differentiate from background levels, is estimated for a SNR of 3 from the chromatograms of the detected samples. Limits of quantification (LOQ), defined as the lowest concentration that the analytical process can quantify from background levels, is estimated for a SNR of 10 from the chromatograms of the detected samples (Zhu et al., J. Chromatogr. A, 2019).

In the supplementary figure below, we show representative UPLC-MS/MS MRM chromatograms of 20E in three different developmental stages, including larval molting stage (third-instar larva to fourth-instar larva, L4-0 h), feeding intermolt stage (fourth-instar larva, L4-24

h) and pupal stage (pupa-24 h) of DBM1Ac-S strain as examples. We can see that the SNR value (i.e. S/N: RMS value in the figure) is only 5.51 with a small peak for L4-0 h, while it is 27.28 with a distinguishable peak for L4-24 h. In contrast, the SNRs are as high as 596.29 and 803.36 with obvious peaks for pupa-24 h and standard solution of 20E, respectively. Although the 20E titer in L4-0 h is higher than LOD but under LOQ, the peak area is too small to be properly used for 20E titer quantification. We also find the same situation in all the larval molting stages of *P. xylostella*. Previously, we ignored the SNR limitation and didn't consider the RMS noise, because SNR is meaningless since the peak signal is distinguishable for 20E titer detection in all the other detected samples except for larval molting stages.

It is worth noting that hormone detection throughout the life cycle of *P. xylostella* is challenging because their body size is much smaller than other extensively studied lepidopteran insects such as *M. sexta* and *B. mori*, especially in the larval stages. One individual is enough for trace amounts of hormone detection for *M. sexta* and *B. mori*, but we need to collect many small *P. xylostella* larvae each time for hormone detection, especially for 20E owing to its low levels. This difficulty is more evident in the larval molting stages since we need to collect enough tiny molting larvae samples during a short time period using a microscope to ensure synchronization. The sample weight is still quite low and just enough for detection. Hence, although the peak area is small in larval molting stages, when we use the calculated whole 20E titer divided by the low sample weight to obtain the standard 20E titer, it will be quite high and this also contributes to the unexpected peak of 20E levels on day 0 of each larval instar.

With a better understanding of the limitations involved in the detection of 20E levels in the larval molting stages, we have revised the manuscript accordingly (lines 239-245) and have redrawn Fig. 5b and Fig. 5d to resolve this problem.

Supplementary Figure. Representative UPLC-MS/MS MRM chromatograms of 20E in the larval molting stage (a), feeding intermolt stage (b) and pupal stage (c) of DBM1Ac-S strain. d The UPLC-MS/MS MRM chromatogram of the standard sample of 20E (5 ng) with the internal control 22S, 23S-homobrassinolide (50 ng). MRM transitions: 20E, 481.1 > 445.2, 481.1 > 371.2; 22S, 23S-homobrassinolide, 495.3 > 127.0; 495.3 > 109.0.

Minor concerns that need attention:

1) In Fig. 5b, they show titers for days 0 and 1 of each larval instar and then connect all the points as if there was only 1 day in each of the first 3 instars and 2 days in the 4th instar. Yet in Fig. 5 C they have points for 0-36 h in L3 and for 0-72 hr in L4. Therefore, the lines should not be connected between the 3rd, 4th and pupa in Fig. 5b.

As pointed out by the reviewer, we indeed missed some developmental time points in the figure, which potentially can cause confusion. To address this concern, additional experiments have been carried out to fill the gaps. Fig. 5b has been redrawn in the revised manuscript. In addition, the time point L4-72 h in Fig. 5c is actually P-0 h, we have also corrected it in Fig. 5c.

2) In the legend to figure 7, line 1134, one should indicate when the hormone was given as follows: "...treatment on day 0 of the 3rd larval instar." In the text describing this figure, they should indicate whether those larvae given 20E molted early to the 4th instar and to the pupa. One would expect that the small size of the resultant pupa is due to a premature molt to the 4th instar but that 4th instar larva is above the threshold size for metamorphosis so goes through a normal length 4th instar. Certainly one injection of 20E would not last into the 4th instar. More discussion is needed for the interpretation of these results.

We have clarified the time at which exogenous hormone was given in the corresponding section of the revised manuscript (legend to Fig. 7). As for the smaller size of the resultant pupa in the 20E-treated group, we attributed this to the 20E-treated third-instar larvae molted early to the fourth-instar and pupal stages. Based on our observation, typical 20E-treated larvae had reduced appetite and food intake before pupation, while the situation was opposite in the methoprene-treated group, in which these larvae had a larger pupal size. In contrast, these observed abnormalities didn't occur in the combined hormone-treated group, which translated into a normal sized pupa. We have discussed this phenomenon in the revised manuscript (lines 323-329).

Reviewer #2 (Remarks to the Author):

The authors performed additional experiments that in my point of view address all reviewers' comments. In particular they performed experiments to define if the higher titers of hormones are cause or effect of the resistance mechanism. The authors show that the increased susceptibility of larvae treated with dsMAP4K4 RNA is reversed by treatment with hormones and that high 20E-hormone titer is genetically linked to Cry1Ac resistance. Although the genetic basis for the high 20E-hormone titer still needs to be define, I believe that this study is a breakthrough in the role of insect hormones in triggering resistance to pathogen infection in insects.

We appreciate the reviewer's positive comments, and we hope our efforts described below can address the remaining concerns.

Minor comments

1. The genetic basis of increase 20E-hormone titer in NIL-R strain still needs to be define. Are there any other examples of insect mutations that cause enhanced hormone titers in insects? What are the phenotypes? Discuss briefly in Discussion section.

We concur with reviewer's assessment. A similar example was documented in a *Drosophila virilis* mutant strain, in which an as yet uncharacterized gene mutation in chromosome 6 elevated the titers of both JH and 20E. These hormone changes mirrored those seen in wild type flies following heat stress and resulted in the same reduced fertility phenotype (decreases in egg production and

oviposition) (Gruntenko et al., *Insect Mol. Biol.*, 2003; Gruntenko and Rauschenbach, *J. Insect Physiol.*, 2008). We have briefly discussed these results in the revised manuscript (lines 457-463).

2. PxAPN3a is slightly down regulated in all strains. If similar genetic basis for resistance is shared in all resistant strain how differences in APN expression of other APN's (e.g. PxAPN13, PxAPN2 etc.) are explained. Additional mutations? Brief discussion of the possible cause could be included in Discussion section.

It is possible that additional factors/mutations could influence PxAPN expression as well. The expression levels of some PxAPN genes, such as *PxAPN13*, were extremely low in larval midgut tissues (Fig. S2). When the Ct values reach the detection limit (~32), the relative expression levels of genes is harder to predict. As for other PxAPN genes, such as *PxAPN2*, that were predominantly expressed in the midgut tissues, the expression differences could reflect the different genetic/epigenetic backgrounds. Given that, we only elected to focus on the four PxAPN genes (*PxAPN1*, *PxAPN3a*, *PxAPN5* and *PxAPN6*) that were mainly expressed in the midgut tissues with significant expression differences between susceptible and all the resistant *P. xylostella* strains. Hence, considering that the other PxAPN genes are not the major common responsive genes involved in Bt resistance in the midgut tissues in *P. xylostella*, we do not feel it is helpful to speculate on any possible role of these other proteins in the Discussion section.

References:

1. Destrez B, Pinel G, Monteau F, Bizec, BL. Development of an LC-MS/MS method for the determination of 20-hydroxyecdysone and its metabolites in calf urine. *Agilent Technologies Technical Note*. 2008, 5989-8879EN. <https://www.gimitec.com/file/5989-8879EN>.
2. Hikiba J, et al. Simultaneous quantification of individual intermediate steroids in silkworm ecdysone biosynthesis by liquid chromatography-tandem mass spectrometry with multiple reaction monitoring. *J Chromatogr B* 915–916, 52–56 (2013).
3. Venne P, Yargeau V, Segura PA. Quantification of ecdysteroids and retinoic acids in whole daphnids by liquid chromatography-triple quadrupole mass spectrometry. *J Chromatogr A* 1438, 57–64 (2016).
4. Wells G, Prest H, Russ IV CW. Why use signal-to-noise as a measure of MS performance when it is often meaningless? *Agilent Technologies Technical Note*. 2011, 5990-8341EN. <https://www.agilent.com/cs/library/technicaloverviews/public/5990-8341EN.pdf>.
5. Zhu Z, Zhang Y, Wang J, Li X, Wang W, Huang Z. Sugaring-out assisted liquid-liquid extraction coupled with high performance liquid chromatography-electrochemical detection for the determination of 17 phenolic compounds in honey. *J Chromatogr A* 1601, 104–114 (2019).
6. Gruntenko NE, Bownes M, Terashima J, Sukhanova MZ, Raushenbach IY. Heat stress affects oogenesis differently in wild-type *Drosophila virilis* and a mutant with altered juvenile hormone and 20-hydroxyecdysone levels. *Insect Mol Biol* 12, 393–404 (2003).
7. Gruntenko NE, Rauschenbach IY. Interplay of JH, 20E and biogenic amines under normal and stress conditions and its effect on reproduction. *J Insect Physiol* 54, 902–908 (2008).

We have carefully revised the manuscript according to the reviewers' suggestions. Hopefully, our efforts and the revised manuscript can ease reviewers' concerns and meet the journal standards.

Please contact me if further modifications are required.

Sincerely Yours

Youjun Zhang

Reviewers' Comments:

Reviewer #1:

Remarks to the Author:

In this revision the authors have met my criticisms by doing more experiments. What they now show is that with resistance to Bt toxin comes a significant change in the endocrine milieu in that the juvenile hormone (JH) titer is higher in the penultimate and final larval instars coupled with a higher ecdysteroid titer at the time of molting in each. Moreover, with the more frequent time points, they show that the developmental time course of the titers of both hormones are in line with those of other lepidopterans. It is really now a lovely piece of work and the findings add a significant new dimension to our understanding of the resistance mechanism. Several minor changes are suggested before publication:

1) The supplementary figure showing the representative chromatograms of 20E and the explanation that they included in their response to the referee should be included in the published paper as it really clarifies their findings in Figure 5. The technique that they have used for ecdysteroid detection is a new one for the insect endocrine field so not many people are familiar with the details and the problems inherent in measuring very low titers.

2) JH II should be written with a space between JH and the Roman numeral II.

3) The data presented in Figure 6b,c and lines 279-290 suggest that the hormones are each acting to change susceptibility through the MAPK cascade of gut enzymes. Yet when the MAPK is knocked down by RNAi, there was no difference in response to 20E alone or to 20E plus JH in terms of either a decrease in enzyme transcription or a decrease in larval mortality (Fig. 6d/e). My interpretation of these latter data is that 20E is not working via the MAPK cascade. I do not understand their conclusion in lines 298-9 about 20E action dominating that of the RNAi. A more complete explanation is needed here.

Reviewer #2:

Remarks to the Author:

The authors address all concerns raised by both reviewers. Specially the enhanced hormone titers observed by reviewer 1

Reviewers' comments:

Reviewer #1 (Remarks to the Author):

In this revision the authors have met my criticisms by doing more experiments. What they now show is that with resistance to Bt toxin comes a significant change in the endocrine milieu in that the juvenile hormone (JH) titer is higher in the penultimate and final larval instars coupled with a higher ecdysteroid titer at the time of molting in each. Moreover, with the more frequent time points, they show that the developmental time course of the titers of both hormones are in line with those of other lepidopterans. It is really now a lovely piece of work and the findings add a significant new dimension to our understanding of the resistance mechanism.

We appreciate the reviewer's positive comments, and we hope our further efforts described below can address the remaining minor concerns.

Several minor changes are suggested before publication:

1) The supplementary figure showing the representative chromatograms of 20E and the explanation that they included in their response to the referee should be included in the published paper as it really clarifies their findings in Figure 5. The technique that they have used for ecdysteroid detection is a new one for the insect endocrine field so not many people are familiar with the details and the problems inherent in measuring very low titers.

We have added this figure as Supplementary Fig. 7 and revised the corresponding Methods section (Lines 877-885) to describe it. Moreover, we have also participated in the transparent peer review scheme as encouraged by the journal, thus, all of the reviewers' comments and our point-by-point responses will be listed as a Peer Review File in the Supplementary information section accompanying the published paper. This will greatly facilitate interested readers to follow the very helpful comments from the reviewer concerning the issues surrounding the accurate measurement and interpretation of hormone titer data.

2) JH II should be written with a space between JH and the Roman numeral II.

We have corrected this formatting error in the whole manuscript and corresponding figures.

3) The data presented in Figure 6b,c and lines 279-290 suggest that the hormones are each acting to change susceptibility through the MAPK cascade of gut enzymes. Yet when the MAPK is knocked down by RNAi, there was no difference in response to 20E alone or to 20E plus JH in terms of either a decrease in enzyme transcription or a decrease in larval mortality (Fig. 6d/e). My interpretation of these latter data is that 20E is not working via the MAPK cascade. I do not understand their conclusion in lines 298-9 about 20E action dominating that of the RNAi. A more complete explanation is needed here.

We also picked up on this particular anomaly, while these data in Fig. 6d/e taken in isolation may suggest no interaction between 20E and the MAPK cascade, the data in Fig 6a/b/c provide strong evidence that it does. We surmise that the effect of 20E in enhancing *MAP4K4* transcription (Fig. 6a) can overcome the effect of RNAi. This is what we meant by 20E action being dominant

over RNAi and so have now modified the corresponding section of the manuscript (lines 342-347) to clarify exactly what we meant.

Reviewer #2 (Remarks to the Author):

The authors address all concerns raised by both reviewers. Specially the enhanced hormone titers observed by reviewer 1.

We thank the reviewer for helping us improve the quality of this study.

We have carefully revised the manuscript according to the reviewer's suggestions. Hopefully, our efforts and the revised manuscript can further address the reviewer's concerns and meet the journal's standards. Thanks very much.

Sincerely Yours

Youjun Zhang
